# Neural Networks and Solomonoff Induction

## Abstract

Solomonoff Induction (SI) is the most powerful universal predictor given unlimited computational resources. Naive SI approximations require running vast amount of programs for extremely long. Here we explore an alternative path to SI consisting in meta-training neural networks on universal data sources. We generate the training data by feeding random programs to Universal Turing Machines (UTMs) and guarantee convergence in the limit to various SI variants (under certain assumptions). Experimentally, we investigate the effect neural network architectures (i.e. LSTMs, Transformers, etc.) and sizes on their performance on algorithmic data, crucial for SI. We test our networks on variable-order Markov sources (VOMS), challenging algorithmic tasks on different levels of the Chomsky hierarchy requiring different memory structures and, finally, on UTM-generated data following our theoretical results. We show that scaling network size always improves performance on all tasks, Transformers outperforming all others, even achieving optimality on VOMS. Promisingly, large Transformers and LSTMs trained on UTM data exhibit transfer to the other domains.

## 1 Introduction

Inductive inference is the process of deriving general rules out of a finite set of concrete examples and using these rules for prediction (Angluin & Smith, 1983). Failed attempts to formalize inductive inference abound (Gabbay et al., 2011; Rathmanner & Hutter, 2011), however, Solomonoff (1964a;b) proposed a universal theory of induction that seems to address all issues that plagued previous methods (Hutter, 2007). Solomonoff Induction (SI) is universal because it considers as hypothesis class the space programs that run in a Universal Turing Machine (UTM) i.e. all computable functions. In addition, SI has an in-built Occam's razor, a bias towards simplicity, quantized/formalized by Kolmogorov complexity (Li et al., 2019), i.e. short programs capture simple explanations. This effectively renders Solomonoff Induction as doing Bayesian inference on program space with a prior favoring shorter program lengths. Remarkably, Solomonoff showed that the posterior over hypotheses (programs on a UTM) converges rapidly to the true sequence-generator $\mu$ (Li & Vitanyi, 1992; Hutter, 2004; Sunehag & Hutter, 2013; Li et al., 2019), with the only requirement that $\mu$ is a computable function. Although incomputable, this p(oste)rior is limit-computable, i.e. SI becomes more accurate as programs are allowed to run for longer. To remedy SI's incomputability, though still intractable, time-limited (finitely computable) approximations were developed such as the Speed Prior (Schmidhuber, 2002; Filan et al., 2016). For more limited model classes, specifically variable-order Markov processes, very efficient Bayes-optimal algorithms such as the Context Tree Weighting (CTW) algorithm exist (Willems et al., 1995; Willems, 1998; Veness et al., 2012). Given the appeal of SI for general intelligence, it is surprising that there is no research in approximating it with modern neural networks.

Recently, Ortega et al. (2019); Mikulik et al. (2020); Genewein et al. (2023) have shown that meta-trained neural networks can learn Bayesian updating, key to Solomonoff induction, in the infinite training limit. This is remarkable since it opens a path for amortizing the powerful Solomonoff prediction (SP) into a neural network architecture, greatly saving computational costs. To make this possible we need two additional components, i.e. universal network architectures and the right data distribution. Most neural network architectures are actually universal in theory (Chen et al., 2017; Stogin et al., 2020; Mali et al., 2023), however, they empirically lose universality under stochastic gradient descent (SGD) training (Deletang et al., 2022).

Unfortunately, attaining easily-trainable universal architectures is still an open problem, and it is not the focus of our paper. Nevertheless, we experiment with existing architectures augmented with a stack or a tape that seem to maintain universality under SGD (Deletang et al., 2022). Orthogonally to the universality issue, specifying the right training distribution also presents a challenge, since the right training data must be consistent and make a universal approximator converge to computable versions of SI.

The aim of this paper is to explore the idea of amortizing SI into a neural network by using synthetic data from UTMs for training. Of course, fully arriving to SI is a futile endeavor (since it is incomputable), however, increasing model sizes, data sizes and computational resources should produce better approximations. First, we theoretically investigate the dataset generation process and training protocol that would make a model converge to SP. We show how properly masking the loss function is critical to converge to a normalized version of SI (Lattimore et al., 2011) that has practical relevance. We make use of a generalized Solomonoff prior (introducing a shorter self-contained proof than (Sterkenburg, 2017)) maintaining the universality property for non-uniform programs. Second, we conduct extensive experiments with RNNs, LSTMs, Transformers etc. on three types of algorithmic data generators with various degrees of complexity and universality, namely variable-order Markov data, on algorithmic tasks at various levels of the Chomsky hierarchy (Deletang et al., 2022) and on algorithmic data generated by UTMs. Our experimental results show that for all architectures, as model size increases, they attain lower cumulative regret on test trajectories.

Our contributions can be summarized as follows.

- We theoretically show how to generate synthetic universal data that make universal approximators converge to SI in the limit, and a self-contained short proof stating that universality is maintained even for non-uniform distributions over programs.

- Our extensive experiments show that increasing architecture size improves performance on all models and tasks. On variable-order Markov sources, large LSTMs and Transformers obtain optimal in-distribution performance, suggesting the capability of implementing Bayesian-mixtures over programs.

- We show how large Transformers trained on UTM data outperform non-trivial Solomonoff estimates and exhibit transfer to our other two tasks, suggesting that the UTM data contains rich enough transferable patterns.

## 2 BACKGROUND

We begin with some terminology for sequential data generating sources. An alphabet is a finite, non-empty set of symbols, denoted by $\mathcal{X}$. A string $x_1 x_2 \ldots x_n \in \mathcal{X}^n$ of length $n$ is denoted by $x_{1:n}$. The prefix $x_{1:j}$ of $x_{1:n}$, $j \leq n$, is denoted by $x_{\leq j}$ or $x_{<j+1}$. The empty string is denoted by $\epsilon$. Our notation also generalizes to out of bounds indices; i.e. given a string $x_{1:n}$ and an integer $m > n$, we define $x_{1:m} := x_{1:n}$ and $x_{n:m} := \epsilon$. The concatenation of two strings $s$ and $r$ is denoted by $sr$.

**Semimeasures.** A semimeasure is a probability measure $P$ over infinite and finite sequences $\mathcal{X}^\infty \cup \mathcal{X}^*$ for some finite alphabet $\mathcal{X}$ assumed to be $\{0, 1\}$ (most statements hold for arbitrary finite $\mathcal{X}$). Let $\mu(x)$ be the probability that an (in)finite sequence *starts* with $x$. While proper distributions satisfy $\sum_{a \in \mathcal{X}} \mu(xa) = \mu(x)$, semimeasures exhibit *probability gaps* and satisfy $\sum_{a \in \mathcal{X}} \mu(xa) \leq \mu(x)$.

**Turing Machines.** A Turing Machine (TM) takes a string of symbols $z$ as an input, and outputs a string of symbols $x$ (after reading $z$ and halting), i.e. $T(z) = x$. For convenience we define the output string at computation step $s$ as $T^s(z) = x$ which may be the empty string $\epsilon$. We adopt similar notation for Universal Turing Machines $U$. Monotone TMs, see Definition 1 below, are special TMs that can incrementally build the output string while incrementally reading the input program, which is a convenient practical property we exploit in our experiments.

**Definition 1** (Monotonicity). *A universal machine $U$ is monotone if for all $p, q, x, y$ with $U(p) = y$ and $U(q) = x$ we have that $\ell(x) \geq \ell(y)$ and $p \sqsubseteq q$ imply $y \sqsubseteq x$, where $p \sqsubseteq q$ means that $p$ is a prefix string of $q$. See Appendix C for a more thorough description.*

**Algorithmic Data Generating Sources and the Chomsky Hierarchy.** An algorithmic data generating source $\mu$ is simply a computable data source by, for example, a TM $T$. There is a natural hierarchy over machines based on their memory structure known as the Chomsky hierarchy

(CH) (Chomsky, 1956), which classifies sequence prediction problems—and associated automata models that solve them—by increasing complexity. There are four levels in the CH, namely, regular, context-free, context-sensitive, and recursively enumerable. Solving problems on each level requires different memory structures such as finite states, stack, finite tape and infinite tape, respectively. We use tasks on different levels of the CH to asses our models. Note that any reasonable approximation to SP would need to sit at the top of the hierarchy.

**Solomonoff Induction.** Given an observed data string $x_{1:n}$ we want to predict the next symbol $x_{n+1}$. Assuming that strings are drawn from an unknown true probability distribution $\mu$, the probability over the next symbol can be predicted with $\mu(x_{n+1}|x_{1:n}) = \mu(x_{1:n+1})/\mu(x_{1:n})$. The best prior for prediction would be to have $\mu$ itself, but this is usually not accessible and an alternative model $\rho$ is us. Restricting the class of priors to the lower semi-computable semimeasures, we can use a single universal semimeasure $M$ for prediction (in place of $\rho$) widely known as the Solomonoff Universal Prior (see definition below). As an intuition, using the Solomonoff prior to predict $x_{n+1}$ using $x_{1:n}$ would be equivalent to doing Bayesian inference on program space by discarding the ones that do not fit the data while assigning higher probability to shorter programs.

**Definition 2** ((Monotone) Solomonoff). *Let $U$ be a universal monotone machine, then the Solomonoff prior is defined as $M(x) := \sum_{p:U(p)=x*} 2^{-\ell(p)}$ with the sum is over all $p \in \{0,1\}^*$, where the output $x*$ is any string that starts with $x$* and *the whole program $p$ has been read by $U$.*

Solomonoff (1964a) showed that SP converges fast if the data is generated by *any* computable probability distribution $\mu$: $\sum_{t=1}^{\infty} \sum_{x_{<t}} \mu(x_{<t}) \sum_{x \in \mathcal{X}} (M(x|x_{<t}) - \mu(x|x_{<t}))^2 \leq K(\mu) \ln 2 < \infty$, where $K(\mu) := \min_p\{\ell(p) : U(p) = \mu\}$ is the Kolmogorov complexity (Li et al., 2019) of the generator $\mu$ (represented as a bitstring). The Solomonoff predictor is essentially the best predictor given a reference UTM. Although the reference machine can play a significant role for short sequences, asymptotically for growing sequences its choice becomes irrelevant.

There exists a normalized version of the Solomonoff prior (among others (Wood et al., 2013)) that is not a semimeasure but a proper measure i.e., properly normalized (see Definition 3 below), has nicer properties when the sequence $x$ contains incomputable sub-sequences (Lattimore et al., 2011) and maintains the convergence properties of the standard Solomonoff prior. This version of Solomonoff is of interest to us because it is more aligned with neural models (that are also properly normalized) and exhibits more efficient sampling when compared to semimeasures.

**Definition 3** (Normalized Solomonoff Prior). *For $a \in \mathcal{X}$, Solomonoff normalization is defined as* $M^{norm}(\epsilon) := 1, \quad M^{norm}(a|x) := \frac{M(xa)}{\sum_{a \in \mathcal{X}} M(xa)} = \frac{M^{norm}(xa)}{M^{norm}(x)}.$

**Meta-Learning.** A parametric model $\pi_\theta$ can be meta-trained by repeating the following steps: 1) sample a task $\tau$ from the task distribution $p(\tau)$, 2) sample a sequence $x_{1:n}$ from $\tau$, 3) train the model $\pi_\theta$ with the log-loss $-\sum_{t=1}^{n} \log \pi_\theta(x_t|x_{<t})$. It can be shown that under mild assumptions, the fully trained $\pi_\theta$ behaves as a Bayes-optimal predictor, i.e. $\pi_\theta(x_t|x_{<t}) \approx \sum_\tau p(\tau|x_{<t})p(x_t|x_{<t}, \tau)$ where $p(x_t|x_{<t}, \tau)$ is the predictive distribution of $\tau$, and $p(\tau|x_{<t})$ is the posterior of $\tau$ (Ortega et al., 2019). If $\mu$ is a proper measure and $D = (x^1, ..., x^J)$ are sequences cut to length $n$ sampled from $\mu$ with empirical distribution $\hat{\mu}(x) = \frac{1}{J} \sum_{y \in D} \llbracket y = x \rrbracket$, then the log-loss $\text{Loss}(\theta) := -\frac{1}{J} \sum_{x \in D} \sum_{t=1}^{\ell(x)} \log \pi_\theta(x_t|x_{<t}) = -\frac{1}{J} \sum_{x \in D} \log \pi_\theta(x) = -\sum_{x \in \mathcal{X}^n} \hat{\mu}(x) \log p_\theta(x)$ is minimized for $\pi_\theta(x) = \hat{\mu}(x)$ provided $\pi_\theta$ can represent $\hat{\mu}$.

## 3 META-LEARNING AS AN APPROXIMATION TO SOLOMONOFF INDUCTION

Next we aim to provide answers to the following questions. First, *how do we generate data that allows to approximate SI?* Second, given that most architectures are trained with a limited sequence-length, *how does this affect the meta-training protocol of neural models?* Third, *can we use different program distributions (making interesting programs more likely) without losing universality?*

### 3.1 THE RIGHT DATASET: ESTIMATING SOLOMONOFF FROM SOLOMONOFF SAMPLES

Our aim here is to define a data generation process such that when used for training our model $\pi_\theta$ (assuming for now universality and essentially infinite capacity), we obtain an approximation to $M$. We consider the incomputable and computable cases. All proofs can be found in the Appendix A.

**Solomonoff Data Generator (incomputable).** Putting uniform random bits on the (read-only) input tape of a monotone UTM $U$ generates a certain distribution $M$ of (in)finite strings on the output tape. This is exactly Solomonoff's universal a-priori distribution $M$ and a semimeasure in the sense above (see Section 2). Sampling from $M$ is trivial; we just described how. It is easy to see that $M$ is equivalent to the more formal Definition 2. The following proposition shows consistency.

**Proposition 4.** *Let $D := (x^1, ..., x^J)$ be $J$ (in)finite sequences sampled from a semimeasure $\mu$ (e.g. $M$). We can estimate $\mu$ as follows:* $\hat{\mu}_D(x) := \frac{1}{|D|} \sum_{y \in D} [\![\ell(y) \geq \ell(x) \ \wedge \ y_{1:\ell(x)} = x]\!] \xrightarrow{w.p.1} \mu(x)$ *for* $|D| \to \infty$.

Unfortunately there are three infinities which prevent us from using $M$ above. There are infinitely many programs, programs may loop forever, and output strings can have infinite length. Therefore, we define the following computable version of the Solomonoff prior.

**Definition 5** (Computable Solomonoff Prior). *Let programs be of length $\leq L$ and stop $U$ after $s$ steps (denoted $U^s$), or if the output reaches length $n$. Then,*

$$M_{s,L,n}(x) := \sum_{p \in \{0,1\}^{\leq L}: U^s(p) = x*} 2^{-\ell(p)} \ \text{ if } \ \ell(x) \leq n \ \text{ and } \ 0 \ \text{ otherwise}$$

*is a computable version of the Solomonoff prior and a semimeasure.*

We can sample $D^J := (x^1, ..., x^J)$ from $M_{s,L,n}$ in the same trivial way as described above for $M$, but now the involved computation is finite. Note that all sampled strings have length $\leq n$, since $M_{s,L,n}(x) := 0$ for $\ell(x) > n$.

**Proposition 6.** *Let now $D^J := (x^1, ..., x^J)$ be samples from the measure $M_{s,L,n}$. Then, $\hat{M}_{D^J}(x) = \frac{1}{J} \sum_{y \in D^J} [\![\ell(y) \geq \ell(x) \ \wedge \ y_{1:\ell(x)} = x]\!] \longrightarrow M_{s,L,n}(x) \ \text{ for } \ J \to \infty$.*

Since $M(x) = \lim_{s,L,n \to \infty} M_{s,L,n}(x) = \sup_{s,L,n} M_{s,L,n}(x)$, we in particular have $\hat{M}_{D^J} \to M$ for $s, L, n, J \to \infty$. Note that $D^J$ depends on $s, L, n$, but this can easily be avoided by choosing $s(j), L(j), n(j)$ to be any functions tending to infinity, and sampling $x^j$ from $M_{s(j),L(j),n(j)}(x)$ for $j = 1, 2, 3, ....$

**Remark 7.** *Although $M_{s,L,n}$ is computable, it still suffers from two inconveniences. First, sampling from it is inefficient because it is a semimeasure and exhibits a probability gap. Second, we need to differentiate whether programs halt or end up in a infinite non-printing loop (to fill the probability gap with "absorbing" tokens when training). We can bypass these inconveniences by estimating instead the normalized (and computable) version of the Solomonoff prior from Definition 3.*

We can estimate the normalized Solomonoff prior, $M_{s,L,n}^{norm}(x)$, by the following.

**Proposition 8.** *Using the definitions from Proposition 6 we have that*

$$\hat{M}_{s,L,n}^{norm}(x_t | x_{<t}) = \frac{\sum_{y \in D^J} [\![\ell(y) \geq t \ \wedge \ y_{1:t} = x_{1:t}]\!]}{\sum_{y \in D^J} [\![\ell(y) \geq t \ \wedge \ y_{<t} = x_{<t}]\!]} \xrightarrow{J \to \infty} M_{s,L,n}^{norm}(x_t | x_{<t})$$

*Then, we can take the product over $t = 1, ..., n$ to obtain $\hat{M}_{s,L,n}^{norm}(x) \to M_{s,L,n}^{norm}(x) \to M^{norm}(x)$.*

**Summary.** Propositions 4, 6 and 8 state that the data generated by the Solomonoff Data Generator and their respective variants (computable and normalized computable) are statistically consistent, and that training on this data would make an estimator converge to their respective Solomonoff version (under realizability and learnability assumptions).

## 3.2 Training Models on Solomonoff Data using Fixed-Sequence Lengths

Most neural model implementations (specially the Transformer model) require sequences of fixed length (say) $n$. The version of $M$ the neural model learns will depend on how we guarantee that all sequences have length $n$. We drop $s, L, n$ from $M_{s,L,n}^{\cdots}$ since what follows holds for infinite as well as finite values. We focus on describing the training protocol that converges to the normalized version of Solomonoff, $M^{norm}$, since it is what we use in our experiments. We recommend readers interested in the standard version of Solomonoff ($M$) to read the Appendix B, where we pad sequences with absorbing token to fill the probability gap (see Semimeasures in Section 2).

**Normalized Solomonoff $M^{norm}$ with neural networks.** To converge to $M^{norm}$, we pad the $x^j$ in $D^J$ to length $n$ with arbitrary symbols from $\mathcal{X}$ and train on them, but we (have to) cut the log-loss short at $\ell(x^j)$. When doing so, the log-loss takes the form (see Appendix B.1 for derivation that uses Proposition 8):

$$\text{Loss}(\theta) \;=\; -\sum_{t=1}^{n}\sum_{x_{<t}}\Big(\sum_{x_t}\hat{M}_{D^J}(x_{1:t})\Big)\Big(\sum_{x_t}\hat{M}^{norm}(x_t|x_{<t})\log\pi_\theta(x_t|x_{<t})\Big) \tag{1}$$

The last bracket and hence the loss is minimized for $\pi_\theta(x_t|x_{<t}) = \hat{M}^{norm}(x_t|x_{<t})$, as desired. By the chain rule this implies that the neural model $\pi_\theta(x)$ converges to $\hat{M}^{norm}(x)$. Note that $\text{Loss}(\theta)$ does *not* depend on the padding of $x^j$, so any padding leads to the same gradient and same solution.

Under the (unrealistic) assumptions that the neural model has the capacity to represent $\hat{M}^{\cdots}$, and the learning algorithm can find the representation, this (tautologically) implies that the neural model distribution $\pi_\theta$ converges to $\hat{\mu} = \hat{M}^{\cdots}$ minimizing (e.g. Eq.(1)). Similarly, if the neural model is trained on $x^j$ sampled from $M^{\cdots}_{s(j),L(j),n}(x)$ for $j = 1, 2, 3, ...$, it converges to $M^{\cdots}_{\infty,\infty,n}$. For a neural model with context length $n$ increasing over time, even $\hat{M}^{\cdots} \to M^{\cdots}_{\infty,\infty,\infty}$ could be possible. Of course there are many practical challenges that need to be surmounted to remotely achieve this.

### 3.3 Solomonoff from Non-Uniform Samples

For practical purposes, sampling from non-uniform (possibly learned) distribution over programs can be advantageous. For our BrainPhoque language (that we use in our experiments later) it increases the yield of 'interesting' programs by a factor of 137 (see Table 3). Below we show this can be done without any concerns on loosing universality.

Let $Q$ be a probability measure on $\mathcal{X}^\infty$, with shorthand $Q(q) := Q(\Gamma_q)$, the $Q$-probability that a sequence starts with $q$, where $\Gamma_q := \{\omega \in \mathcal{X}^\infty : q \sqsubseteq \omega\} = q\mathcal{X}^\infty$. We define the *generalized Solomonoff semimeasure* as

$$M_T^Q(x) \;:=\; \sum_{q:T(q)=x*} Q(q) \quad\text{with special case}\quad M_U(x) \;:=\; \sum_{q:U(q)=x*} 2^{-\ell(q)}$$

for a universal TM $T = U$ and unbiased coin flips $Q(q) = 2^{-\ell(q)}$. $M_U$ is strongly universal in the sense that it is a Bayesian mixture over all lower semi-computable semimeasures (Wood et al., 2011). Next, we show to we show that under very mild conditions on $Q$, $M_U^Q$ is also universal. This finding is similar to (Sterkenburg, 2017), but our independently discovered proof is shorter and more self-contained.

**Theorem 9** (Universality of generalized Solomonoff semimeasures). *$M_U^Q(x)$ is strongly universal, provided $Q$ is a computable measure and $Q(q) > 0 \,\forall q \in \mathcal{X}^*$ and $Q(q_{1:n}) \to 0$ for $n \to \infty$. More precisely, for all universal monotone TM $U$ and all $Q$ with the above properties, there exists a universal MTM $V$ (as constructed in the proof) s.th. $M_U^Q(x) = M_V(x) \,\forall x$. Proof in Appendix C.*

**Note on the assumptions above.** We assumed infinite number of data points and universality (and learnablity) of the approximator, which are difficult to obtain in practice and diminish the relevance of inductive biases of neural models. For finite data, however, inductive biases are crucial for generalization. In addition, easily-trainable universal neural models is key for (practically) learning data on the highest level of the Chomsky hierarchy, i.e. UTM generated data. We leave out of the scope of the paper the remaining theoretical work on the effect of the inductive bias and universality of neural models and focus on providing experimental evidence of neural network performance on UTM generated data in the next section.

## 4 Experimental Methodology

Our experiments consist of evaluating neural networks with different architectures and sizes trained on three types of algorithmically generated data described next.

**Variable-order Markov Sources (VOMS).** A Markov model of order $k$ sequentially assigns probabilities to a string of characters by looking, at step $t$ in the sequence, at the suffix string from $t - k$

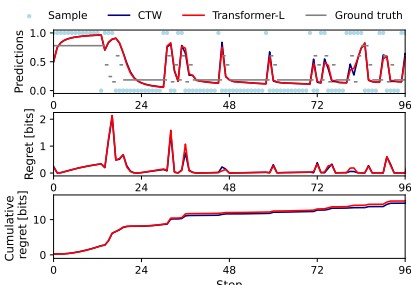 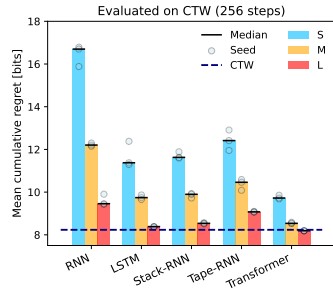 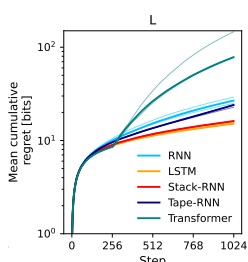

Figure 1: Evaluation on data from the **variable-order Markov source**. **Left:** Example sequence and predictions of Transformer-L (red) and Bayes-optimal CTW predictor (blue), below we show instantaneous and cumulative regret w.r.t. the ground-truth. **Middle:** Mean cumulative regret over 6k sequences (length 256, max. CTW tree depth 24, in-distribution) for different networks (3 seeds) and sizes (S, M, L). Larger models perform better for all architectures, and the Transformer-L and LSTM-L match the optimal CTW predictor. **Right:** Length generalization (1024 steps). LSTMs generalize to longer length, whereas Transformers do not.

to $t$. This suffix is used to lookup the model parameters to make a prediction of the next character. A VOMS is a Markov model where the value of $k$ can vary depending on the suffix, and makes its prediction using a suffix tree. We consider binary VOMS where an efficient Bayes-optimal predictor exists: the Context Tree Weighting (CTW) predictor (Willems et al., 1995; 1997). We use the CTW predictor as a baseline to get a better sense of how close are our models to Bayes-optimality. CTW predictor is only universal w.r.t. $n$-Markov sources, and thus not universal w.r.t. all computable functions like SI. See Appendix D.2 for more intuition about VOMS and information on how do we generate the data and how to compute the CTW Bayes-optimal predictor.

**Chomsky Hierarchy (CH) Tasks.** We take the 15 algorithmic tasks (e.g. arithmetic, reversing strings etc.) from Deletang et al. (2022) lying on different levels of the Chomsky hierarchy (see Appendix D.3 for a description of all tasks). In contrast to Deletang et al. (2022) who train on *individual* tasks, we are interested in the more challenging setting where we train on all tasks *simultaneously*. To do this we make sure that all tasks use the same alphabet $\mathcal{X}$ (effectively expanding the alphabet of tasks with small alphabet size). In addition, we do not consider transduction as in Deletang et al. (2022) but sequence prediction, thus we concatenate inputs and outputs with additional delimiter tokens i.e. for $\{(x_i \in \mathcal{X}, y_i \in \mathcal{X})\}_{i=1}^I$ and delimiters ',' and ';', we construct sequences of the form $z := (x_1, y_1; x_2, y_2; \dots x_n, y_n; \dots)$. When evaluating our models we only account the regret (and accuracy) on the output symbols (masking every other symbol) because inputs are usually random and non-informative of whether the task has been solved. Denoting $\mathcal{O}_z$ the set of outputs time-indices, we compute accuracy for trajectory $z$ as $A(z) := \frac{1}{|\mathcal{O}_z|} \sum_{t \in \mathcal{O}_z} [\arg\max_y \pi_\theta(y|z_{<t}) = z_t]$. See Appendix D.3 for more details.

**Universal Turing Machine Source.** This data source is based on all the details on Sections 3.1 and 3.2. We generate random programs (that can effectively encode any structured sequence generation process) and run them in our UTM to get the outputs. In principle a program could generate the image of a cow, a chess program, or the books of Shakespeare, but of course, these programs are extremely unlikely to be sampled (see Figure 5 in the Appendix for exemplary outputs). As a choice of UTM we constructed a variant of the brainf*ck UTM (Müller, 1993), which we call Brain-Phoque, mainly to help with the sampling process and to ensure that all sampled programs are valid. We also use an alphabet size $|\mathcal{X}| = 17$ equal to the Chomsky tasks to enable evaluation between tasks. BrainPhoque has a single working tape and a write-only output tape. It has 7 instructions to move the working tape pointer (WTP), de/increment the value under the WTP (the *datum*), perform jumps and append the datum to the output. All programs are valid, and imbalanced brackets are simply skipped. While we slightly change the program distribution, this is not an issue according to Theorem 9. Programs are sampled and run for $s = 1000$ steps with 200 memory cells, with a maximum output length of $n = 256$ symbols. Ideally, the Solomonoff predictor should be the optimal baseline to compare to but it is uncomputable and intractable. Hence, we calculate a (rather loose, but non-trivial) upper bound on the log-loss incurred by the Solomonoff predictor by using

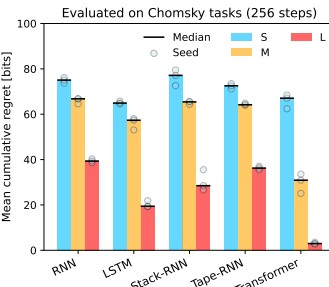
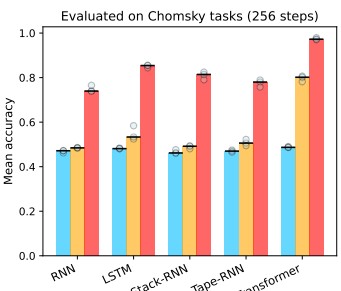
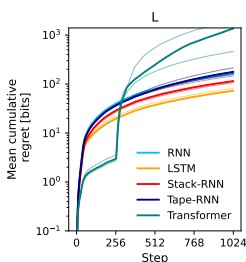

Figure 2: Evaluation on 6k sequences from the **Chomsky hierarchy tasks** (400 per task). As the model size increases, cumulative regret (**Left**) and accuracy (**Middle**) improve across all architectures. Overall, the Transformer-L achieves the best performance by a margin. **Right:** Length generalization (1024 steps). Detailed results per task are in Figure 7 on the Appendix.

the prior probability of shortened programs (by removing unnecessary brackets or self-canceling instructions) that generate the outputs. See Appendix E for a full description of BrainPhoque and our procedure to sample programs and outputs.

**Neural Predictors.** Our neural models $\pi_\theta$ sequentially observe symbols $x_{<t}$ from the data generating source and predict the next-symbol probabilities $\pi_\theta(\cdot|x_{<t})$. We train our models on 256-length trajectories using the log-loss $\text{Loss}(\theta) := -\frac{1}{n}\sum_{t=1}^{n}\log \pi_\theta(x_t|x_{<t})$ and mini-batch (size 128) stochastic gradient descent with the ADAM optimizer (Kingma & Ba, 2014) (learning rate $10^{-4}$). On the UTM data source, we cut the log-loss on short sequences to approximate the normalized version of SI (see Section 3.2). We evaluate the following architectures: RNNs, LSTMs, Stack-RNNs, Tape-RNNs and Transformers. We note that, Stack-RNNs (Joulin & Mikolov, 2015) and Tape-RNNs (Deletang et al., 2022) are RNNs augmented with a stack and tape memory, respectively, that can be used to store and manipulate symbols. We consider three model sizes (S, M and L) for each architecture by increasing the width and depth simultaneously. We train 3 initialization seeds per model variation for 500K SGD iterations. We expect that larger networks have an increased capacity to represent longer computational traces and more parallel throughput and thus better performance. The reason for including memory-augmented models is to check whether these networks can meaningfully use their external memory to predict better. We expect this to be difficult to achieve in practice. See Appendix D.1 for all the architecture details.

**Evaluation procedure.** Our main evaluation metric is the *expected instantaneous regret*, $R_{\pi\mu}(t) := \mathbb{E}_{x_t\sim\mu}\left[\log\mu(x_t \mid x_{<t}) - \log\pi(x_t \mid x_{<t})\right]$ (at time $t$), and *cumulative expected regret*, $R_{\pi\mu}^T := \sum_{t=1}^{T} R_{\pi\mu}(t)$, where $\pi$ is the model and $\mu$ the ground-truth source. We evaluate our neural models on 6k sequences of length 256, which we refer as *in-distribution* (same length as used for training) and of length 1024, referred as *out-of-distribution*.

## 5   RESULTS

**Variable-order Markov Source (VOMS) Results.** In Figure 1 (Left) we show an example trajectory from VOMS data-source of length 256 with the true samples (blue dots), ground truth (gray), Transformer-L (red) and CTW (blue) predictions. As we can see the predictions of the CTW predictor and the Transformer-L are aligned, suggesting it is implementing a Bayesian mixture over programs/trees. In the second and third panels the instantaneous regret and the cumulative regret also match. Figure 1 (Middle) shows the cumulative regret of all the neural predictors evaluated in-distribution. First, we observe that as model size increases (from S, M, to L) the cumulative regret decreases. In addition, the best model is the Transformer-L, which remarkably achieves optimal performance, whereas the worst models are the RNNs and the Tape-RNNs. The latter model probably could not properly leverage its external memory successfully. Note also how LSTM-L achieves close to optimal performance. On the Right we show out-of-distribution performance showing how transformers fail on length-generalization. To better understand where our models struggle, we show in the Appendix F, Figures 6c and 6d, the cumulative regret averaged across seeds and trajectories

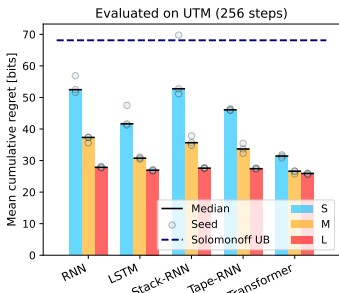
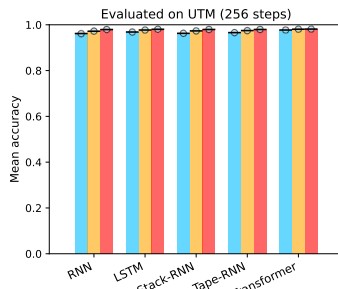
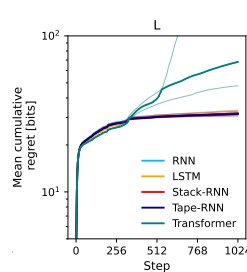

Figure 3: Evaluation on the **UTM data generator** with 6k sequences. **Left:** The larger the architecture the lower the cumulative regret. We see better performance than the non-trivial baseline Solomonoff Upper Bound (UB). **Middle:** The mean accuracy on UTM data shows the models can quickly learn UTM patterns. **Right:** Length generalization (1024 steps). Detailed results per program length are in Figure 8.

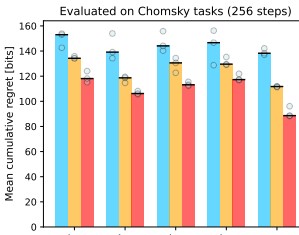
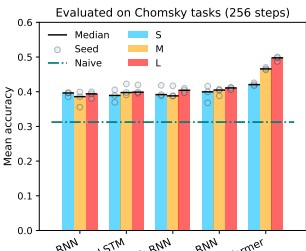
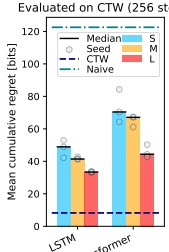
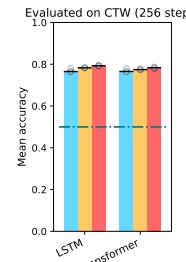

Figure 4: Transfer learning from **UTM-trained models** on 3k trajectories. Mean cumulative regret (**Left**) and accuracy (**Middle-Left**) of neural models trained on UTM data evaluated against the tasks of the Chosmky hierarchy. We observe a small increase in accuracy (transfer) from the Transformer models. Transfer to CTW is shown in the right two panels: **Middle-Right:** mean cumulative regret, **Right:** mean accuracy; 'Naive' is a random uniform predictor.

sampled from different CTW tree depths and context lengths. We can see as the size grows from S to L, the cumulative regret above the CTW optimal is more or less uniform for all tree-depths and higher on short context-lengths.

**Chomsky Hierarchy Results.** In Figure 2 (Left) we show the in-distribution performance of all our models trained on the Chomsky hierarchy tasks by means of cumulative regret and accuracy. Overall, the Transformer-L achieves the best performance by a margin. On the Right we show the length-generalization capabilities of models, showing how Transformers fail to generalize to longer lengths. In the Appendix (Figure 7) we show the results for each task individually.

**Universal Turing Machine Results.** Figure 3 (Left) shows the mean cumulative regret on the UTM task with the (loose) Solomonoff Upper Bound (UB) as a non-trivial baseline (see Section 4 for its description). In the Middle we show the accuracy, and as can be seen all models achieve fairly good accuracy. The main reason for this is that naively sampling from our UTM produces fairly simple programs (see example UTM trajectories in appendix Figure 5). In general, larger architectures attain lower cumulative regret and all models show lower regret than the Solomonoff upper bound. Note that the neural models do not have access to the underlying program that generate the output. Interestingly, in Figure 8 (in the Appendix) we show the cumulative regret against program length and observe that the longer the underlying program the higher the cumulative regret of our models since it is more difficult to approximate them. Remarkably, in Figure 4 we see that the Transformer networks trained on UTM data exhibit the most transfer to the Chomsky tasks and, LSTMs transfer the most to the VOMS task (compare to the 'naive' random predictor). For the VOMS, we retrained the LSTM and Transformer models with the BrainPhoque UTM setting the alphabet size to 2 matching our VOMS task to enable comparison. All transfer results suggest that UTM data contains enough transferable patterns for these tasks.

## 6    DISCUSSION AND CONCLUSIONS

**Large Language Models and Solomonoff Induction.**  The last few years the ML community has witnessed the training of enormous models on massive quantities of diverse data (Kenton & Toutanova, 2019; Hoffmann et al., 2022). This trend is in line with the premise of our paper, i.e. to achieve increasingly universal models one needs large architectures and large quantities of diverse data. Large Language Models (LLMs) have been shown to have impressive in-context learning capabilities (Kenton & Toutanova, 2019; Chowdhery et al., 2022). LLMs pretrained on long-range coherent documents can learn new tasks from a few examples by inferring a shared latent concept (Xie et al., 2022; Wang et al., 2023). They can do so because in-context learning does implicit Bayesian inference (in line with our CTW experiments) and builds world representations and algorithms (Li et al., 2023a;b) (necessary to perform SI). In fact, one could argue that the impressive in-context generalization capabilities of LLMs is a sign of a rough approximation of Solomonoff induction. The advantage of pre-trained LLMs compared to our method (of training on UTM data) is that LLM data (books, code, online conversations etc.) is generated by humans, and thus very well aligned with the tasks we (humans) want to solve; whereas our UTMs assign low probability to human tasks. However, mixing our data generation processes with human data might be advantageous towards more general intelligence.

**Learning the UTM.** Theorem 9 of our paper (and (Sterkenburg, 2017)) opens the path for modifying/learning the program distribution of a UTM while maintaining the universality property. In fact, if we want to arrive at a practical approach that solve human problems this is probably necessary, since the naive uniform prior over programs on our UTMs generated only relatively simple data. Interestingly, learning a UTM aligned to problems of interest is also the goal of Sunehag & Hutter (2014). Once having a good UTM, we could use it as a good synthetic data generator to improve our models. This is the idea of data-augmentation that has been so successful in the machine learning field (Perez & Wang, 2017; Lemley et al., 2017; Kataoka et al., 2020). In future work, equipped with our Theorem 9, we plan to explore in depth how to modify the sampling process from UTMs to produce more interesting, more complex or more useful outputs.

**Increasingly Universal Architectures.** By definition, the output of the function $U^s(p)$ (using program $p$) requires at maximum $s$ computational steps. Approximating $M_{s,L,n}$ would naively require wide networks (to represent many programs in parallel) of $s$-depth and context length $n$. The efficiency of this representation would depend on whether computational patterns can be reused. Transformers seem to exhibit such a feature where by means of "shortcuts" they can represent all automata of length $T$ in $O(\log T)$-depth (Liu et al., 2023). An alternative way to increase the amount of serial computations is with chain-of-thought (Wei et al., 2022) (see Hahn & Goyal (2023) for theoretical results). When data is limited, inductive biases are important for generalization. Luckily it seems neural networks have an implicit inductive bias towards simple functions at initialization (Dingle et al., 2018; Valle-Perez et al., 2018; Mingard et al., 2023) compatible with Kolmogorov complexity, which is greatly convenient when trying to approximate SI in the finite-data regime.

**Limitations.** Given the empirical nature of our results, we cannot guarantee, due to training for finite time, that our architectures mimic the universality of Solomonoff (although this becomes more likely as training samples increase). Similarly, we note universality does not help with the fact that Solomonoff Induction is uncomputable/undecidable and one would need infinite time to exactly match Solomonoff in the limit. Our theoretical results establish that good approximations to Solomonoff induction are obtainable, in principle, via meta-training; whereas our empirical results show that is possible to make practical progress in that direction, though many questions remain open, e.g., how to construct efficient relevant universal datasets for meta-learning, and how to obtain easily-trainable universal architectures.

**Conclusion.** We aimed at using meta-learning as driving force to approximate Solomonoff Induction. For this we had to carefully specify the data generation process and the training loss so that the convergence to various versions of Solomonoff predictors is attained in the limit. Our experiments on the three different algorithmic data-sources tell similar stories: as model size increases, performance increases. Remarkably, networks trained on the UTM data-source exhibit some transfer to the other domains. We believe that we can improve LLM training by scaling our approach to generate synthetic data-generation using UTMs and mixing it with existing large datasets.

**Reproducibility Statement.** On the theory side, we wrote all proofs in the Appendix. For data generation, we fully described the variable-order Markov sources in the Appendix; we used the open-source repository https://github.com/google-deepmind/neural_networks_chomsky_hierarchy for the Chomsky tasks and fully described our UTM in the Appendix. We used the same architectures as Deletang et al. (2022) (which can be found in the same open-source repository) with modifications described in the Appendix. For training our models we used JAX https://github.com/google/jax.

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

## 7 APPENDIX

## A SOLOMONOFF SAMPLES

**Sampling from semimeasures.** We can sample strings from a semimeasure $\mu$ as follows: Start with the empty string $x = \epsilon$.
With probability $\mu(a|x) := \mu(xa)/\mu(x)$ extend $x \leftarrow xa$ for $a \in \mathcal{X}$. Repeat.
With probability $1 - \sum_{a \in \mathcal{X}} \mu(a|x)$ return $x$.

Let $D := (x^1, ..., x^J)$ be $J$ (in)finite sequences sampled from $\mu$. If we only have these samples, we can estimate $\mu$ as follows:

$$\hat{\mu}_D(x) := \frac{1}{|D|} \sum_{y \in D} [\![\ell(y) \geq \ell(x) \wedge y_{1:\ell(x)} = x]\!] \overset{w.p.1}{\longrightarrow} \mu(x) \quad \text{for} \quad |D| \to \infty \tag{2}$$

*Proof:* Let $D_x := (y \in D : \ell(y) \geq \ell(x) \wedge y_{1:\ell(y)} = x)$ be the elements in $D$ that start with $x$. Since $x^j$ are sampled i.i.d. from $\mu$, the law of large numbers implies $|D_x|/|D| \to \mu(x)$ for $J \to \infty$. $\quad\square$

**Limit normalization.** A simple way of normalization is

$$\widetilde{M}_{s,L,n}(x_{1:t}) := \frac{\sum_{x_{t+1:n}} M_{s,L,n}(x_{1:n})}{\sum_{x_{1:n}} M_{s,L,n}(x_{1:n})} \quad \text{for} \quad t \leq n \quad \text{and} \quad 0 \quad \text{else}$$

This is a proper measure for sequences up to length $n$. Sampling from it is equivalent to sampling from $M_{s,L,n}$ but discarding all sequences shorter than $n$. Let $\widetilde{D} := (x^j \in D^J : \ell(x^j) \geq n)$. Then

$$\hat{\widetilde{M}}_{\widetilde{D}}(x) = \frac{1}{|\widetilde{D}|} \sum_{y \in \widetilde{D}} [\![y_{1:\ell(x)} = x]\!] \longrightarrow M(x) \quad \text{for} \quad s, L, n, J \to \infty$$

*Proof:* First, $|\widetilde{D}|/|D|$ is the relative fraction of sequences that have length $n$, and $\sum_{x_{1:n}} M_{s,L,n}(x_{1:n})$ is the probability that a sequence has length $n$, hence the former converges to the latter for $J \to \infty$. Second,

$$\hat{\widetilde{M}}_{\widetilde{D}}(x_{1:n}) = \frac{1}{|\widetilde{D}|} \sum_{y \in \widetilde{D}} [\![y_{1:\ell(x)} = x_{1:n}]\!] = \frac{|D|}{|\widetilde{D}|} \frac{1}{|D|} \sum_{y \in D} [\![\ell(y) \geq n \wedge y_{1:\ell(x)} = x_{1:n}]\!]$$

$$= \frac{|D|}{|\widetilde{D}|} \hat{M}_{D^J}(x_{1:n}) \overset{J \to \infty}{\longrightarrow} \frac{M_{s,L,n}(x_{1:n})}{\sum_{x_{1:n}} M_{s,L,n}(x_{1:n})} = \widetilde{M}_{s,L,n}(x_{1:n})$$

Third, take the sum $\sum_{x_{t+1:t}}$ on both sides, and finally the limit $s, L, n \to \infty$ and set $x = x_{1:t}$. $\quad\square$

A disadvantage of this normalization scheme is that the probability of a sequence $x$ depends on $n$ even if $\ell(x) < n$, while $M_{s,L,n}(x)$ and $M_{...}^{norm}(x)$ below are essentially independent of $n$.

**Proposition 4.** *Let* $D := (x^1, ..., x^J)$ *be $J$ (in)finite sequences sampled from a semimeasure $\mu$ (e.g. $M$). We can estimate $\mu$ as follows:* $\hat{\mu}_D(x) := \frac{1}{|D|} \sum_{y \in D} [\![\ell(y) \geq \ell(x) \wedge y_{1:\ell(x)} = x]\!] \overset{w.p.1}{\longrightarrow} \mu(x)$ *for* $|D| \to \infty$.

*Proof:* Let $D_x := (y \in D : \ell(y) \geq \ell(x) \wedge y_{1:\ell(y)} = x)$ be the elements in $D$ that start with $x$. Since $x^j$ are sampled i.i.d. from $\mu$, the law of large numbers implies $|D_x|/|D| \to \mu(x)$ for $J \to \infty$. □

**Proposition 6.** *Let now $D^J := (x^1, ..., x^J)$ be samples from the measure $M_{s,L,n}$. Then, $\hat{M}_{D^J}(x) = \frac{1}{J} \sum_{y \in D^J} [\![\ell(y) \geq \ell(x) \wedge y_{1:\ell(x)} = x]\!] \longrightarrow M_{s,L,n}(x)$ for $J \to \infty$.*

*Proof:* It follows directly from Proposition 4.

**Proposition 8.** *Using the definitions from Proposition 6 we have that*

$$\hat{M}_{s,L,n}^{norm}(x_t | x_{<t}) = \frac{\sum_{y \in D^J} [\![\ell(y) \geq t \wedge y_{1:t} = x_{1:t}]\!]}{\sum_{y \in D^J} [\![\ell(y) \geq t \wedge y_{<t} = x_{<t}]\!]} \overset{J \to \infty}{\longrightarrow} M_{s,L,n}^{norm}(x_t | x_{<t})$$

*Then, we can take the product over $t = 1, ..., n$ to obtain $\hat{M}_{s,L,n}^{norm}(x) \to M_{s,L,n}^{norm}(x) \to M^{norm}(x)$.*

*Proof:* For $x = x_{<t}$ and $a = x_t$, we have

$$\sum_{a \in \mathcal{X}} \hat{M}_{D^J}(xa) = \frac{1}{J} \sum_a \sum_{y \in D^J} [\![\ell(y) \geq \ell(xa) \wedge y_{1:\ell(xa)} = xa]\!]$$

$$= \frac{1}{J} \sum_{y \in D^J} [\![\ell(y) \geq t \wedge \exists a : y_{1:t} = xa]\!]$$

$$\text{hence} \quad \hat{M}_{s,L,n}^{norm}(a|x) = \frac{\hat{M}_{D^J}(xa)}{\sum_a \hat{M}_{D^J}(xa)} \overset{J \to \infty}{\longrightarrow} \frac{M_{s,L,n}(ax)}{\sum_a M_{s,L,n}(ax)} = M_{s,L,n}^{norm}(a|x) \quad (3)$$

□

## B  TRAINING WITH LLMS

**Using Transformer LLMs for estimating $M$.**  Most Transformer implementations require sequences of fixed length (say) $n$. We can mimic shorter sequences by introducing a special absorbing symbol $\perp \notin \mathcal{X}$, and pad all sequences $x^j$ shorter than $n$ with $\perp$s. We train the Transformer on these (padded) sequences with the log-loss. Under the (unrealistic) assumptions that the Transformer has the capacity to represent $\hat{M}_{...}$, and the learning algorithm can find the representation, this (tautologically) implies that the Transformer distribution converges to $\hat{M}_{...}$. Similarly if the Transformer is trained on $x^j$ sampled from $M_{s(j),L(j),n}(x)$ for $j = 1, 2, 3, ...$, it converges to $M_{\infty,\infty,n}$. For a Transformer with context length $n$ increasing over time, even $\hat{M}_{...} \to M$ could be possible. To guarantee normalized probabilities when learning $\widetilde{M}_{...}$ and $M_{...}^{norm}$, we do *not* introduce a $\perp$-padding symbol. For $\widetilde{M}_{...}$ we train on $\widetilde{D}$ which doesn't require padding. For training towards $M_{...}^{norm}$, we pad the $x^j$ in $D^J$ to length $n$ with arbitrary symbols from $\mathcal{X}$ and train on that, but we (have to) cut the log-loss short $-\sum_{t=1}^{\ell(x)} \log(\text{LLM}(x_t | x_{<t}))$, i.e. $\ell(x)$ rather than $n$, so as to make the loss hence gradient hence minimum independent of the arbitrary padding.

**Limit-normalized $\widetilde{M}$.**  This is the easiest case: $\widetilde{D}$ removes strings shorter than $n$ from $D^J$ (sampled from $M$), so $\widetilde{D}$ has distribution $\widetilde{M}$, hence for $D = \widetilde{D}$, the log-loss is minimized by $p_\theta = \hat{\widetilde{M}}$, i.e. training on $\widetilde{D}$ makes $p_\theta$ converge to $\hat{\widetilde{M}}$ (under the stated assumptions).

**Unnormalized $M$.**  For this case we need to augment the (token) alphabet $\mathcal{X}$ with some (absorbing) padding symbol $\perp$: Let $D_\perp$ be all $x \in D^J$ but padded with some $\perp$ to length $n$. We can extend $M : \mathcal{X}^* \to [0;1]$ to $M_\perp : \mathcal{X}^* \cup \{\perp\} \to [0;1]$ by

$$\begin{aligned} M_\perp(x) &:= M(x) & \text{for all} && x \in \mathcal{X}^* \\ M_\perp(x\perp^t) &:= M(x) - \textstyle\sum_{a \in \mathcal{X}} M(xa) & \text{for all} && x \in \mathcal{X}^* \text{ and } t \geq 1 \\ M_\perp(x) &:= 0 & \text{for all} && x \notin \mathcal{X}^*\{\perp\}^* \end{aligned}$$

It is easy to see that $D_\perp$ has distribution $M_\perp$, hence for $D = D_\perp$, the log-loss is minimized by $p_\theta = \hat{M}_\perp$. Since $\hat{M}_\perp(x)$ restricted to $x \in \mathcal{X}^*$ is just $\hat{M}(x)$, training on $D_\perp$ makes $p_\theta(x)$ converge to

$\hat{M}(x)$ for $x \in \mathcal{X}^*$. Though it is possible to train neural models that would converge in the limit to the standard (computable) Solomonoff prior, we focus on the normalized version due to Remark 7. *Training variation:* Note that for $M$, the Transformer is trained to predict $x\perp$ if $\ell(x) < n$. If $\ell(x) < n$ is due to the time limit $s$ in $U^s$, it is preferable to *not* train the Transformer to predict $\perp$ after $x$, since for $s \to \infty$, which we are ultimately interested in, $x$ may be extended with proper symbols from $\mathcal{X}$. One way to achieve this is to cut the log-loss (only) in this case at $t = \ell(x)$ similar to $M^{norm}$ below to not reward the Transformer for predicting $\perp$.

### B.1 Normalized Solomonoff Loss

Here is the derivation of the loss.

$$
\begin{aligned}
\text{Loss}(\theta) &:= -\frac{1}{J} \sum_{x \in D^J} \log p_\theta(x) = -\frac{1}{J} \sum_{x \in D^J} \sum_{t=1}^{\ell(x)} \log p_\theta(x_t | x_{<t}) \\
&= -\frac{1}{J} \sum_{t=1}^{n} \sum_{x \in D^J \wedge \ell(x) \geq t} \log p_\theta(x_t | x_{<t}) = -\sum_{t=1}^{n} \sum_{x_{1:t}} \hat{M}_{D^J}(x_{1:t}) \log p_\theta(x_t | x_{<t}) \\
&= -\sum_{t=1}^{n} \sum_{x_{<t}} \Big( \sum_{x_t} \hat{M}_{D^J}(x_{1:t}) \Big) \Big( \sum_{x_t} \hat{M}^{norm}(x_t | x_{<t}) \log p_\theta(x_t | x_{<t}) \Big)
\end{aligned}
$$

where the last equality follows from (3).

## C  Generalized Solomonoff Semimeasure

**Streaming functions.**  A streaming function $\varphi$ takes a growing input sequence and produces a growing output sequence. In general, input and output may grow unboundedly or stay finite. Formally, $\varphi : \mathcal{X}^\# \to \mathcal{X}^\#$, where $\mathcal{X}^\# := \mathcal{X}^\infty \cup \mathcal{X}^*$. In principle input and output alphabet could be different, but for simplicity we assume that all sequences are binary, i.e. $\mathcal{X} = \{0, 1\}$. For $\varphi$ to qualify as a streaming function, we need to ensure that extending the input only extends and does not modify the output. Formally, we say that

$$\varphi \text{ is monotone} \quad \text{iff} \quad [\forall q \sqsubseteq p : \varphi(q) \sqsubseteq \varphi(p)]$$

where $q \sqsubseteq p$ means that $q$ is a prefix of $p$ i.e. $\exists r \in \mathcal{X}^\# : qr = p$, and $\sqsubset$ denotes strict prefix $r \neq \epsilon$. $p$ is $\varphi$-minimal for $x$ if $\exists r : \phi(p) = xr$ and $\forall r \forall q \sqsubset p : \phi(q) \neq xr$. We will denote this by $\varphi(p) = x*$. $p$ is the shortest program outputting a string starting with $x$.

**Monotone Turing Machines (MTM).**  A Monotone Turing machine $T$ is a Turing machine with left-to-right read-only input tape, left-to-right write-only output tape, and some bidirectional work tape. The function $\varphi_T$ it computes is defined as follows: At any point in time after writing the output symbol but before moving the output head and after moving the input head but before reading the new cell content, if $p$ is the content left of the current input tape head, and $x$ is the content of the output tape up to the current output tape head, then $\varphi_T(p) := x$. It is easy to see that $\varphi_T$ is monotone. We abbreviate $T(p) = \varphi_T(p)$. There exist (so called optimal) universal MTM $U$ that can emulate any other MTM via $U(i'q) = T_i(q)$, where $T_1, T_2, ...$ is an effective enumeration of all MTMs and $i'$ a prefix encoding of $i$ (Hutter, 2004; Li et al., 2019).

### C.1 Proof of Theorem 9

**Theorem 9** (Universality of generalized Solomonoff semimeasures).  *$M_U^Q(x)$ is strongly universal, provided $Q$ is a computable measure and $Q(q) > 0 \ \forall q \in \mathcal{X}^*$ and $Q(q_{1:n}) \to 0$ for $n \to \infty$. More precisely, for all universal monotone TM $U$ and all $Q$ with the above properties, there exists a universal MTM $V$ (as constructed in the proof) s.th. $M_U^Q(x) = M_V(x) \ \forall x$. Proof in Appendix C.*

We can effectively sample from any computable $Q$ if we have access to infinitely many fair coin flips. The conditions on $Q$ ensure that the entropy of $Q$ is infinite, and stays infinite even when conditioned on any $q \in \mathcal{X}^*$. This also allows the reverse: Converting a sample from $Q$ into infinitely

many uniform random bits. Forward and backward conversion can be achieved sample-efficiently via (bijective) arithmetic (de)coding. This forms the basis of the proof below. The condition of $Q$ being a proper measure rather than just being a semimeasure is also necessary: For instance, for $Q(q) = 4^{-\ell(q)}$, on a Bernoulli$(\frac{1}{2})$ sequence $x_{1:\infty}$, $M_U(x_t|x_{<t}) \to \frac{1}{2}$ as it should, one can show that $M_U^Q(x_t|x_{<t}) < \frac{1}{3}$ for infinitely many $t$ (w.p.1).

*Proof. (sketch)* Let $0.q_{1:\infty} \in [0;1]$ be the real number with binary expansion $q_{1:\infty}$. With this identification, $Q$ can be regarded as a probability measure over $[0;1]$. Let $F : [0;1] \to [0;1]$ be its cumulative distribution function, which can explicitly be represented as $F(0.q_{1:\infty}) = \sum_{t:q_t=1} Q(\Gamma_{q_{<t}0})$, since $[0; 0.q_{1:\infty}) = \biguplus_{t:q_t=1} 0.\Gamma_{q_{<t}0}$, where $0.\Gamma_q = [0.q0^\infty; 0.q1^\infty)$ and $\biguplus$ denotes disjoint union. Now assumption $Q(q) > 0 \; \forall q \in \mathcal{X}^*$ implies that $F$ is strictly increasing, and assumption $Q(q_{1:n}) \to 0$ implies that $F$ is continuous. Since $F(0) = 0$ and $F(1) = 1$, this implies that $F$ is a bijection. Let $0.p_{1:\infty} = F(0.q_{1:\infty})$ and $0.q_{1:\infty} = F^{-1}(0.p_{1:\infty})$. [1]. Further for some finite prefix $q \sqsubset q_{1:\infty}$, we partition the interval

$$[0.p_{1:\infty}^0; 0.p_{1:\infty}^1) := [F(0.q0^\infty); F(0.q1^\infty)) =: \biguplus_{p \in \Phi(q)} 0.\Gamma_p$$

into a minimal set of binary intervals $0.\Gamma_p$, where $\Phi(q)$ is a minimal prefix free set in the sense that for any $p$, at most one of $p, p0, p1$ is in $\Phi(q)$. An explicit representation is

$$\Phi(q) := \{p_{<t}^0 1 : t > t_0 \wedge p_t^0 = 0\} \dot\cup \{p_{<t}^1 0 : t > t_0 \wedge p_t^1 = 1\}$$

where $t_0$ is the first $t$ for which $p_t^0 \neq p_t^1$. Now we plug

$$Q(q) = F(0.q1^\infty) - F(0.q0^\infty) = \sum_{p \in \Phi(q)} |0.\Gamma_p| = \sum_{p \in \Phi(q)} 2^{-\ell(p)} \quad \text{into}$$

$$M_U^Q(x) \equiv \sum_{q:U(q)=x*} Q(q) = \sum_{q:U(q)=x*} \sum_{p \in \Phi(q)} 2^{-\ell(p)} = \sum_{p:V(p)=x*} 2^{-\ell(p)} = M_V(x)$$

where $V(p) := U(q)$ for the maximal $q$ such that $p \in \Phi(q)$. The maximal $q$ is unique, since $\Phi(q) \cap \Phi(q') = \{\}$ if $q \not\sqsubseteq q'$ and $q' \not\sqsubseteq q$, and finite since $F$ is continuous.

It remains to show that $V$ is universal. Let $p^i$ be such that $0.\Gamma_{p^i} \subseteq [F(0.i'0^\infty); F(0.i'1^\infty))$. The choice doesn't matter as long as it is a computable function of $i$, but shorter is "better". This choice ensures that $F^{-1}(0.p^i*) = 0.i'...$ whatever the continuation $*$ is. Now let $F(q_{1:\infty})_{\text{tail}} := F(q_{1:\infty})_{\ell(p^i)+1:\infty} = p_{\ell(p^i)+1:\infty}$ if $q_{1:\infty}$ starts with $i'$, and arbitrary, e.g. $F(q_{1:\infty})$, otherwise. Let $T$ be a MTM with $T(q_{1:\infty}) := U_0(F(q_{1:\infty})_{\text{tail}})$ for some universal MTM $U_0$. By Kleene's 2nd recursion theorem (Sipser, 2012, Chp.6), there exists an $i$ such that $T_i(q) = T(i'q) \; \forall q$. Let $\dot{k} := \ell(i') + 1$ and $\dot{\ell} := \ell(p^i) + 1$ and $q_{<k} := i'$, hence $p_{<\ell} = p^i$. Now $V(p_{1:\infty}) = U(q_{1:\infty})$ implies

$$V(p^i p_{\dot\ell:\infty}) = U(i'q_{\dot k:\infty}) = T_i(q_{\dot k:\infty}) = T(i'q_{\dot k:\infty}) = U_0(F(i'q_{\dot k:\infty})_{\text{tail}}) = U_0(p_{\dot\ell:\infty})$$

hence $V$ is universal, which concludes the proof. $\square$

**Practical universal streaming functions.** Turing machines are impractical and writing a program for a universal streaming function is another layer of indirection which is best to avoid. Programming languages are already universal machines. We can define a conversion of real programs from/to binary strings and prepend it to the input stream. When sampling input streams $q_{1:\infty}$ we convert the beginning into a program of the desired programming language, and feed it the tail as input stream.

# D   EXPERIMENT METHODOLOGY DETAILS

## D.1   ARCHITECTURE DETAILS

**RNN.** A vanilla multi-layer RNN (Elman, 1990) with hidden sizes and multi-layer perceptron (MLP) before and after the RNN layers as described in Table 1.

---

[1]Note that $p_{1:m}$ is uniformly distributed and is (for some $m$) essentially the arithmetic encoding of $q_{1:n}$ with one caveat: The mapping from sequences to reals conflates $0.q10^\infty = 0.q01^\infty$. Since the set of all conflated sequences has probability 0, (under $Q$ as well as Bernoulli$(\frac{1}{2})$), any error introduced due to this conflation has no effect on the distribution $M_U^Q(x)$.

Table 1: Architectures

| RNN and LSTMs | S | M | L |
|---|---|---|---|
| RNN Hidden size | 16 | 32 | 128 |
| Number of RNN layers | 1 | 2 | 3 |
| MLP before RNN layers | (16,) | (32, 32) | (128, 128, 128) |
| MLP after RNN layers | (16,) | (32, 32) | (128, 128, 128) |
| **Transformer SINCOS** | | | |
| Embedding dimension | 16 | 64 | 256 |
| Number of heads | 2 | 4 | 4 |
| Number of layers | 2 | 4 | 6 |

**Stack-RNN.** A multi-layer RNN controller with hidden sizes and MLP exactly the same as the RNN and LSTMs on Table 1 with access to a differentiable stack (Joulin & Mikolov, 2015). The controller can perform any linear combination of PUSH, POP, and NO-OP on the stack of size according to Table 1, with action weights given by a softmax over a linear readout of the RNN output. Each cell of the stack contains a real vector of dimension 6 and the stack size is 64 for all (S, M and L) sizes.

**Tape-RNN.** A multi-layer RNN controller with hidden sizes according to the Table 1 with access to a differentiable tape, inspired by the Baby-NTM architecture (Suzgun et al., 2019). The controller can perform any linear combination of WRITE-RIGHT, WRITE-LEFT, WRITE-STAY, JUMP-LEFT, and JUMP-RIGHT on the tape, with action weights given by a softmax. The actions correspond to: writing at the current position and moving to the right (WRITE-RIGHT), writing at the current position and moving to the left (WRITE-LEFT), writing at the current position (WRITE-STAY), jumping $\ell$ steps to the right without writing (JUMP-RIGHT), where $\ell$ is the length of the input, and jumping $\ell$ steps to the left without writing (JUMP-LEFT). As in the Stack-RNN, each cell of the tape contains a real vector of dimension 6 and the tape size is 64 for all (S, M and L) sizes.

**LSTM.** A multi-layer LSTM (Hochreiter & Schmidhuber, 1997) of hidden sizes according to Table 1.

**Transformer decoder.** A vanilla Transformer decoder (Vaswani et al., 2017). See Table 1 for the embedding dimension, number of heads and number of layers for each model size (S, M and L). Each layer is composed of an attention layer, two dense layers, and a layer normalization. We add a residual connections as in the original architecture (Vaswani et al., 2017). We consider the standard sin/cos (Vaswani et al., 2017) positional encoding.

### D.2 CTW

Below is an ultra-compact introduction to (sampling from) CTW (Willems et al., 1995; 1997). For more explanations, details, discussion, and derivations, see (Catt et al., 2024, Chp.4).

**A variable-order Markov process.** is a probability distribution over (binary) sequences $x_1, x_2, x_3, ...$ with the following property: Let $S \subset \{0, 1\}^*$ be a complete suffix-free set of strings (a reversed prefix-free code) which can equivalently be viewed as a perfect binary tree. Then $p(x_t = 0 | x_{<t}; S, \Theta_S) := \theta_s$ if (the unique) context of $x_t$ is $s = x_{t-\ell(s):t-1} \in S$, and $\Theta_S := (\theta_s \in [0;1] : s \in S)$. We arbitrarily define $x_t = 0$ for $t \leq 0$.

**Intuition about Variable-order Markov sources** VOMS considers data generated from tree structures. For example, given the binary tree

```
        Root
     0/       \1
 Leaf_0       Node
```

```
           0/          \1
      Leaf_10         Leaf_11
```

and given the history of data "011" (where 0 is the first observed datum and 1 is the last one) the next sample uses $\text{Leaf}_{11}$ (because the last two data points in history were 11) to draw the next datum using a sample from a Beta distribution with parameter $\text{Leaf}_{11}$. Say we sample a 0, thus history is then transformed into "0110" and $\text{Leaf}_{10}$ will be used to sample the next datum (because now the last two datapoints that conform to a leaf are "10"), and so forth. This way of generating data is very general and can produce many interesting patterns ranging from simple regular patterns like 01010101 or more complex ones that can have stochastic samples in it. Larger trees can encode very complex patterns indeed.

**Sampling from CTW.** Context Tree Weighting (CTW) is a Bayesian mixture over all variable-order Markov sources of maximal order $D \in \mathbb{N}_0$, i.e. over all trees $S$ of maximal depth $D$ and all $\theta_s \in [0;1]$ for all $s \in S$. The CTW distribution is obtained as follows: We start with an empty (unfrozen) $S = \{\epsilon\}$. Recursively, for each unfrozen $s \in S$ with $\ell(s) < D$, with probability $^1/_2$ we freeze $s$; with probability $^1/_2$ we split $S \leftarrow S \setminus \{s\} \cup \{0s, 1s\}$ until all $s \in S$ are frozen or $\ell(s) = D$. Then we sample $\theta_s$ from $\text{Beta}(^1/_2, ^1/_2)$ for all $s \in S$. Finally for $t = 1, 2, 3, ...$ we sample $x_t$ from $p(x_t | x_{<t}; S, \Theta_S)$.

**Computing CTW.** The CTW probability $P_{\text{CTW}}(x_{1:n})$ can be calculated as follows: Let $a_s := |\{t \in \{1, ..., n\} : x_t = 0 \wedge x_{t-\ell(s):t-1} = s\}|$ count the number of $x_t = 0$ immediately preceded by context $s \in \{0,1\}^*$, and similarly $b_s := |\{t : x_t = 1 \wedge x_{t-\ell(s):t-1} = s\}|$. Let $x_{1:n}^s \in \{0,1\}^{a_s+b_s}$ be the subsequence of $x_t$'s that have context $s$. For given $\theta_s$ for $s \in S$, $x_{1:n}^s$ is i.i.d. (Bernoulli($1 - \theta_s$)). Hence for $\theta_s \sim \text{Beta}(^1/_2, ^1/_2)$, $P(x_{1:n}^s | s \in S) = P_{\text{KT}}(a_s, b_s) := \int_0^1 \theta_s^{a_s}(1 - \theta_s)^{b_s} \text{Beta}(^1/_2, ^1/_2)(\theta_s) d\theta_s$. If $s \notin S$, we split $x_{1:n}^s$ into $x_{1:n}^{0s}$ and $x_{1:n}^{1s}$. By construction of $S$, a tentative $s \in S$ gets replaced by $0s$ and $1s$ with 50% probability, recursively, hence $P_{\text{CTW}}(x_{1:n}^s) = \frac{1}{2}P_{\text{KT}}(a_s, b_s) + \frac{1}{2}P_{\text{CTW}}(x_{1:n}^{0s})P_{\text{CTW}}(x_{1:n}^{1s})$ terminating with $P_{\text{CTW}}(x_{1:n}^s) = P_{\text{KT}}(a_s, b_s)$ when $\ell(s) = D$. This completes the definition of $P_{\text{CTW}}(x_{1:n}) \equiv P_{\text{CTW}}(x_{1:n}^\epsilon)$. Efficient $O(nD)$ algorithms for computing $P_{\text{CTW}}(x_{1:n})$ (and updating $n \rightarrow n+1$ in time $O(D)$) and non-recursive definitions can be found in Catt et al. (2024, Chp.4).

**Distributions of Trees.** A tree has depth $\leq d$ if either it is the empty tree or if both its subtrees have depth $< d$. Therefore the probability of sampling a tree of depth $\leq d$ is $F(d) = \frac{1}{2} + \frac{1}{2}F(d-1)^2$, with $F(0) = \frac{1}{2}$. Therefore the probability of sampling a tree of depth $d$ is $P(d) = F(d) - F(d-1)$ for $d < D$ and $P(D) = 1 - F(D-1)$. The theoretical curve ($P(0) = \frac{1}{2}$, $P(1) = \frac{1}{8}$, $P(2) = \frac{9}{128}$,...) is plotted in Fig. 6a together with the empirical distribution. More meaningful is probably the expected number of leaf nodes at each level $d$. Since each node at level $d$ is replaced with prob. $\frac{1}{2}$ by two nodes at level $d+1$, the expected number of leaf nodes $E(d)$ is the same at all levels $d < D$. Since $E(0) = \frac{1}{2}$, we have $E(d) = \frac{1}{2}$ for all $d < D$ and $E(D) = 1$, hence the total expected number of leaf nodes is $E_+ = \frac{1}{2}D + 1$. While this doesn't sound much, it ensures that for $N = 10'000$ samples, we uniformly test $5'000$ contexts for each length $d < D$. We can get some control over the distribution of trees by splitting nodes at level $d$ with probability $\alpha_d \in [0;1]$ instead of $\frac{1}{2}$. In this case, $E(d) = 2\alpha_0 \cdot ... \cdot 2\alpha_{d-1}(1 - \alpha_d)$ for $d < D$. So for $\alpha_d > \frac{1}{2}$ we can create trees of size exponential in $D$, and (within limits) any desired depth distribution.

### D.3 CHOMSKY

## E   UTMs: BRAINF*CK AND BRAINPHOQUE

Our BrainPhoque (BP) UTM produces program evaluation traces that are equivalent to those of brainf*ck (BF) programs (Müller, 1993) (see also $\mathcal{P}''$ (Böhm, 1964)), but the programs are written slightly differently: they are even less human-readable but have better properties when sampling programs.

We start by giving a quick overview of the BF machine, then we explain why we need a slightly different machine, and we explain its construction next. Finally we explain how to shorten sampled

Table 2: Table taken from (Deletang et al., 2022). Tasks with their level in the Chomsky hierarchy and example input/output pairs. The † denotes permutation-invariant tasks; the ⋆ denotes counting tasks; the ∘ denotes tasks that require a nondeterministic controller; and the × denotes tasks that require superlinear running time in terms of the input length.

| Level | Name | Example Input | Example Output |
|---|---|---|---|
| Regular (R) | Even Pairs | $aabba$ | True |
| | Modular Arithmetic (Simple) | $1 + 2 - 4$ | 4 |
| | Parity Check† | $aaabba$ | True |
| | Cycle Navigation† | $011210$ | 2 |
| Deterministic context-free (DCF) | Stack Manipulation | $abbaa$ POP PUSH $a$ POP | $abba$ |
| | Reverse String | $aabba$ | $abbaa$ |
| | Modular Arithmetic | $-(1-2) \cdot (4 - 3 \cdot (-2))$ | 0 |
| | Solve Equation∘ | $-(x-2) \cdot (4 - 3 \cdot (-2))$ | 1 |
| Context-sensitive (CS) | Duplicate String | $abaab$ | $abaababaab$ |
| | Missing Duplicate | $10011021$ | 0 |
| | Odds First | $aaabaa$ | $aaaaba$ |
| | Binary Addition | $10010 + 101$ | 10111 |
| | Binary Multiplication× | $10010 * 101$ | 1001000 |
| | Compute Sqrt | $100010$ | 110 |
| | Bucket Sort†⋆ | $421302214$ | $011222344$ |

programs and calculate an upper bound on the log-loss that Solomonoff induction based on this BF UTM would incur.

See Figure 5 for some sample programs and outputs.

### E.1 BRAINF*CK

BF is one of the smallest and simplest Turing-complete programming languages. It features a read-only input tape, a working tape, and a write-only output tape. These tapes are assumed infinite but for practical purposes they are usually fixed at a finite and constant length and initialized with 0.[2] Each tape cell can contain an integer between 0 and 255 — we say that the alphabet size is 256, but in the experiments we use an alphabet size of 17. Each tape has a pointer, and when a program is being evaluated there is also an instruction pointer.

For simplicity, the pointer of the working tape is called WTP, and the value at the WTP is called *datum*, which is an integer.

BF uses 8 instructions <>+-[],. which are:

- < and > decrement and increment the WTP, modulo the length of the tape.
- + and - increment and decrement the datum, modulo the alphabet size.
- [ is a conditional jump: if the datum is 0, the instruction pointer jumps to the corresponding (matching) ].
- ] is an unconditional jump to the corresponding [.[3]
- , copies the number under the reading tape pointer into the datum cell, and increments the reading pointer.
- . copies the datum to the output tape at the output pointer and increments the output pointer.

In this paper we do not use an input tape, so we do not use the , instruction.

When evaluating a program, the instruction pointer is initially on the first instruction, the output tape is empty, and the working tape is filled with zeros. Then the instruction under the instruction

---

[2]The tape could also grow on request, but this tends to slow down program evaluation.

[3]For efficiency reasons the instruction ] is usually defined to jump to the matching [ if the datum is non-zero. We stick to a unconditional jump for simplicity reasons.

pointer is evaluated according to the above rules, and the instruction pointer is moved to the right. Evaluation terminates when the number of evaluated instructions reaches a given limit, or when the number of output symbols reaches a given limit.

For a sequence of instructions `A[B]C`, where `A`, `B` and `C` are sequences of (well-balanced) instructions, we call `B` the *body* of the block and `C` the *continuation* of the block.

### E.2   BRAINPHOQUE: SIMULTANEOUS GENERATION AND EVALUATION

We want to sample arbitrary BF programs and evaluate them for $T$ steps each. To maximize computation efficiency of the sampling and running process, programs containing unbalanced parentheses are made valid, in particular by skipping any additional `]`.

Since we want to approximate *normalized* Solomonoff induction 3, we can make a few simplifications. In particular, programs do not need to halt explicitly, which removes the need for a halting symbol and behaviour.[4] Hence we consider that *all* programs are infinite, but that at most $T$ instructions are evaluated. The difficulty with BF programs is that the evaluated instructions can be at arbitrary locations on the program tape, since large blocks `[...]` may be entirely skipped, complicating both the sampling process and

This can be fixed by generating BF programs as trees, where branching on opening brackets `[`: The left branch corresponds to the body of the block (and terminates with a `]`), while the right branch corresponds to the continuation of the block. When encountering an opening bracket for the first time during evaluation, which branch is evaluated next depends on the datum. Hence, to avoid generating both branches, we need to generate the program *as it is being evaluated*: when sampling and evaluating a `[`, if the datum is 0 we follow the right branch and start sampling the continuation without having to sample the body (for now); conversely, if the datum is not zero, we follow the left branch and start sampling and evaluating the continuation. If the same opening bracket is later evaluated again with a different datum value, the other branch may be generated and evaluated.

Our implementation of program generation and evaluation in BrainPhoque uses one growing array for the program, one jump table, and one stack for yet-unmatched open brackets.

If the instruction pointer is at the end of the program, a new instruction among `+-<>[].` is sampled; if it is `[` and the datum is 0, it is changed to `{`. The new instruction is appended to the program, and is then evaluated. If the new instruction is `[`, the next instruction to be sample (and appended to the program) is the beginning of the body of the block, but if instead the new instruction is `{`, the next instruction to be sampled (and appended to the program) is the continuation of the body. At this point the jump table does not yet need to be updated — since the next instruction to evaluate is also the next instruction in location. The jump table is updated to keep track of where the continuations and bodies are located in the program. If the instruction pointer eventually comes back for a second time of an opening bracket `[` (resp. `{`) and the datum is now 0 (resp. not 0), the continuation (resp. body) of the block must now be sampled and appended to the program; and now the jump table must be updated accordingly.

The stack of unmatched brackets is updated only when the body of a block is being generated.

Some properties of BrainPhoque:

- If a program is run for $t+k$ steps, it behaves the same on the first $t$ steps for all values of $k$.[5] In particular, unmatched opening brackets behave the same whether they will be matched or not.

- Program generation (sampling) only requires a single growing-only array. A tree structure is not required. This is the reason for having the additional `{` instruction, which makes it clear — once evaluated the second time — whether the body or the continuation has already been generated.

---

[4]The halting behaviour can be recovered by ending programs with a particular infinite loop such as `[]+[]` (which loops whether the datum is zero or not), and terminate the evaluation (instead of looping forever) upon evaluating this sequence.

[5]While this is an obviously desirable property, it is also easy to overlook.

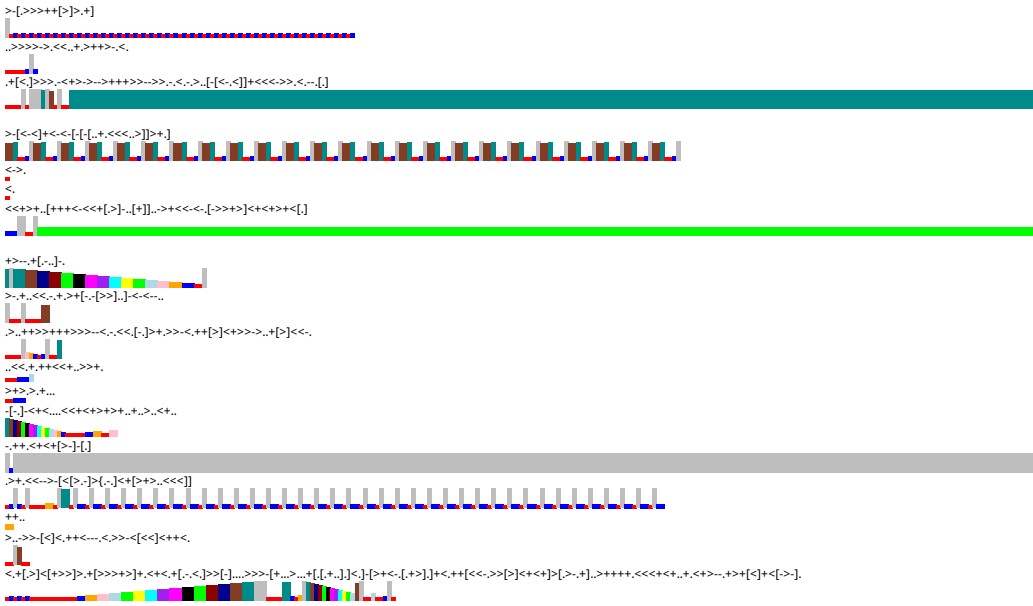

Figure 5: Some BrainPhoque programs and their corresponding outputs (truncated at 256 symbols). The smallest bars (in red) correspond to the value 0, and the largest bars (in gray) correspond to value 16. The programs have been reduced after evaluation by removing a set of unnecessary instructions. Most of the generated outputs are regular, and only about 1 in 5000 sampled programs exhibits non-regular patterns. But see Table 3 for a way to improve these numbers and generate more interesting and complex sequences.

- If the instruction pointer is at cell $n$, then all instructions to the left of $n$ have been evaluated at least once. If this is the first evaluation of cell $n$, then no instruction to the right of $n$ have been evaluated yet.

### E.3 SOLOMONOFF LOG-LOSS UPPER BOUND AND SHORTENING PROGRAMS

We tried to provide a meaningful upper bound for the loss of Solomonoff induction for Figure 3, but this is far from easy. See Section 4 for context. As mentioned there, to calculate a more meaningful upper bound, we shorten programs by recursively removing unnecessary open brackets and closing brackets that are unmatched, as well as all self-cancelling pairs of instructions (+-, -+, <>,><). Moreover, we remove all instructions of the program that have been evaluated for the first time after the last evaluation of a print . instruction (since they do not participate in producing the output. This procedure often reduces programs by a third. Programs that do not output anything are thus reduced to the empty program (probability 1).

If $q$ is a sampled program, then $\tilde{q}$ is the corresponding shortened program. We calculate an upper bound on the loss of the Solomonoff predictor, with U = BrainPhoque, on a set of sampled programs $\hat{Q} = (q^1, \ldots, q^J)$ and corresponding outputs $(U(q^1)_{1:256}, \ldots, U(q^J)_{1:256})$,

$$\text{Loss}(M_U, \hat{Q}) = \sum_{q \in \hat{Q}} - \log \sum_{p:U(p)_{1:256}=U(q)_{1:256}} 7^{-\ell(p)} \leq \sum_{q \in \hat{Q}} - \log 7^{-\ell(\tilde{q})} = \log(7) \sum_{q \in \hat{Q}} \ell(\tilde{q}) \quad (4)$$

since the program alphabet is not binary but has 7 instructions. Unfortunately, even after reduction this bound is still quite loose, but improving this bound meaningfully would likely require a much larger amount of computation.

Markov chain order 0

| Ctx. | < | > | + | - | [ | ] | . | Freq. |
|---|---|---|---|---|---|---|---|---|
|  | .14 | .14 | .14 | .15 | .08 | .08 | .27 | 1.000 |

Markov chain order 1

| Ctx. | < | > | + | - | [ | ] | . | Freq. |
|---|---|---|---|---|---|---|---|---|
| _ | .19 | .19 | .19 | .20 | .00 | .00 | .23 | .018 |
| + | .18 | .17 | .17 | .00 | .14 | .07 | .27 | .141 |
| - | .17 | .17 | .00 | .17 | .13 | .08 | .28 | .144 |
| . | .14 | .15 | .15 | .15 | .07 | .09 | .25 | .272 |
| < | .17 | .00 | .19 | .18 | .06 | .10 | .30 | .139 |
| > | .00 | .18 | .17 | .19 | .05 | .11 | .30 | .140 |
| [ | .15 | .14 | .15 | .15 | .12 | .01 | .28 | .082 |
| ] | .15 | .17 | .16 | .17 | .01 | .09 | .25 | .064 |

Markov chain order 2

| Ctx. | < | > | + | - | [ | ] | . | Freq. |
|---|---|---|---|---|---|---|---|---|
| __ | .19 | .19 | .19 | .20 | .00 | .00 | .23 | .018 |
| _+ | .22 | .24 | .19 | .00 | .11 | .00 | .24 | .004 |
| _- | .15 | .27 | .00 | .21 | .13 | .00 | .24 | .004 |
| _. | .17 | .22 | .21 | .17 | .00 | .00 | .23 | .004 |
| _< | .23 | .00 | .28 | .23 | .00 | .00 | .26 | .004 |
| _> | .00 | .18 | .17 | .26 | .00 | .00 | .39 | .003 |
| ++ | .19 | .17 | .17 | .00 | .15 | .06 | .26 | .023 |
| +. | .15 | .13 | .14 | .13 | .11 | .08 | .26 | .039 |
| +< | .18 | .00 | .19 | .19 | .05 | .09 | .30 | .025 |
| +> | .00 | .19 | .19 | .18 | .06 | .09 | .29 | .024 |
| +[ | .16 | .14 | .15 | .14 | .12 | .01 | .28 | .020 |
| +] | .12 | .17 | .17 | .16 | .02 | .11 | .25 | .006 |
| -- | .17 | .17 | .00 | .17 | .15 | .07 | .27 | .024 |
| -. | .14 | .15 | .13 | .14 | .11 | .09 | .24 | .040 |
| -< | .17 | .00 | .20 | .19 | .07 | .08 | .29 | .025 |
| -> | .00 | .20 | .18 | .20 | .05 | .08 | .29 | .024 |
| -[ | .16 | .14 | .14 | .15 | .13 | .01 | .27 | .019 |
| -] | .18 | .18 | .18 | .19 | .01 | .06 | .20 | .007 |
| .+ | .17 | .16 | .16 | .00 | .12 | .09 | .30 | .041 |
| .- | .17 | .17 | .00 | .17 | .11 | .09 | .29 | .040 |
| .. | .14 | .15 | .16 | .15 | .06 | .11 | .23 | .066 |
| .< | .18 | .00 | .19 | .17 | .05 | .10 | .31 | .039 |
| .> | .00 | .16 | .18 | .18 | .05 | .12 | .31 | .041 |
| .[ | .14 | .15 | .14 | .17 | .11 | .01 | .28 | .019 |
| .] | .16 | .17 | .16 | .16 | .01 | .08 | .26 | .019 |
| <+ | .19 | .18 | .16 | .00 | .16 | .04 | .27 | .026 |
| <- | .21 | .16 | .00 | .18 | .13 | .06 | .26 | .025 |
| <. | .14 | .16 | .14 | .15 | .03 | .11 | .27 | .042 |
| << | .18 | .00 | .19 | .19 | .05 | .09 | .30 | .024 |
| <[ | .14 | .16 | .17 | .16 | .11 | .01 | .25 | .008 |
| <] | .14 | .17 | .16 | .18 | .01 | .11 | .23 | .012 |
| >+ | .18 | .16 | .19 | .00 | .14 | .06 | .27 | .025 |
| >- | .17 | .18 | .00 | .17 | .15 | .05 | .28 | .027 |
| >. | .14 | .14 | .16 | .16 | .05 | .10 | .25 | .042 |
| >> | .00 | .18 | .18 | .21 | .05 | .09 | .29 | .025 |
| >[ | .15 | .15 | .15 | .17 | .11 | .01 | .26 | .007 |
| >] | .16 | .15 | .15 | .18 | .01 | .09 | .26 | .013 |
| [+ | .17 | .14 | .14 | .00 | .12 | .13 | .30 | .012 |
| [- | .17 | .11 | .00 | .16 | .09 | .15 | .32 | .013 |
| [. | .15 | .17 | .16 | .14 | .10 | .01 | .27 | .023 |
| [< | .15 | .00 | .12 | .13 | .07 | .21 | .32 | .012 |
| [> | .00 | .15 | .10 | .14 | .07 | .21 | .33 | .012 |
| [[ | .14 | .13 | .17 | .14 | .09 | .00 | .33 | .010 |
| [] | .11 | .11 | .33 | .11 | .00 | .06 | .28 | .001 |
| ]+ | .16 | .15 | .16 | .00 | .14 | .15 | .24 | .010 |
| ]- | .13 | .17 | .00 | .15 | .15 | .17 | .23 | .011 |
| ]. | .15 | .19 | .16 | .19 | .01 | .08 | .22 | .016 |
| ]< | .17 | .00 | .17 | .16 | .07 | .14 | .29 | .009 |
| ]> | .00 | .19 | .19 | .19 | .05 | .12 | .26 | .011 |
| ][ | .13 | .13 | .10 | .27 | .10 | .00 | .27 | .001 |
| ]] | .13 | .20 | .17 | .17 | .00 | .09 | .24 | .005 |

Table 3: **Pre-trained BP program sampling probabilities** Instead of sampling programs uniformly, we can sample them w.r.t. any probability distribution $Q$ that satisfies Theorem 9. We initially sampled programs uniformly and filtered out 'boring' sequences. Then we trained $Q$ via cross-entropy to mimic the distribution of 'interesting' sequences. We used a 2nd-order Markov process as a model for $Q$. While uniform sampling resulted in only 0.02% interesting sequences, sampling from $Q$ increased it to 2.5%, a 137-fold improvement. The table on the left shows the 0th, 1st, and 2nd order Markov processes $Q(p_t)$, $Q(p_t|p_{t-1})$, and $Q(p_t|p_{t-2}p_{t-1})$ from which BP programs are sampled, for $p. \in \{<>+-[]\{.\}$, but where results for [ and { have been merged. Each row corresponds to a context (none or $p_{t-1}$ or $p_{t-2}p_{t-1}$). We also included $Q(p_1|p_0{:=}\_)$ and $Q(p_1|p_{-1}p_0{:=}\_\_)$. The entries in each column correspond to the sampling probability of $p_t$ in the corresponding row-context. Training on interesting sequences has led to a non-uniform distribution $Q$. Universality is preserved for any $k$-order Markov process, provided all transition probabilities are non-zero. The probability $Q(.)$ of outputting a symbol has nearly doubled from 0.14 to 0.27 on average, while the probability of loop brackets ([, ]) reduced to 0.07 each on average. The marginal probabilities $Q(<) \approx Q(>) \approx Q(+) \approx Q(-) \approx 1/7$ have not changed much, but many of the conditional ones have. Certain combination of instructions are now blocked: For instance +- and -+ and <> and >< have probability close to 0, since they cancel each other and hence are redundant. Some triples such as ][- and <+ and >- and others are enhanced.

Caveat: We did not have time to retrain our NN models on these newly generated sequences (experiments are still running). But since the statistics is improved, we expect the results in Figures 3 and 4 to improve or at least not deteriorate.

# F    ADDITIONAL RESULTS DETAILS

Below we show additional results of the experiments on the VOMS (Figure 6), the Chomsky tasks (Figure 7) and UTM source (Figures 8 and 9). Finally, on Figure 10 we show further details of the length generalization analysis.

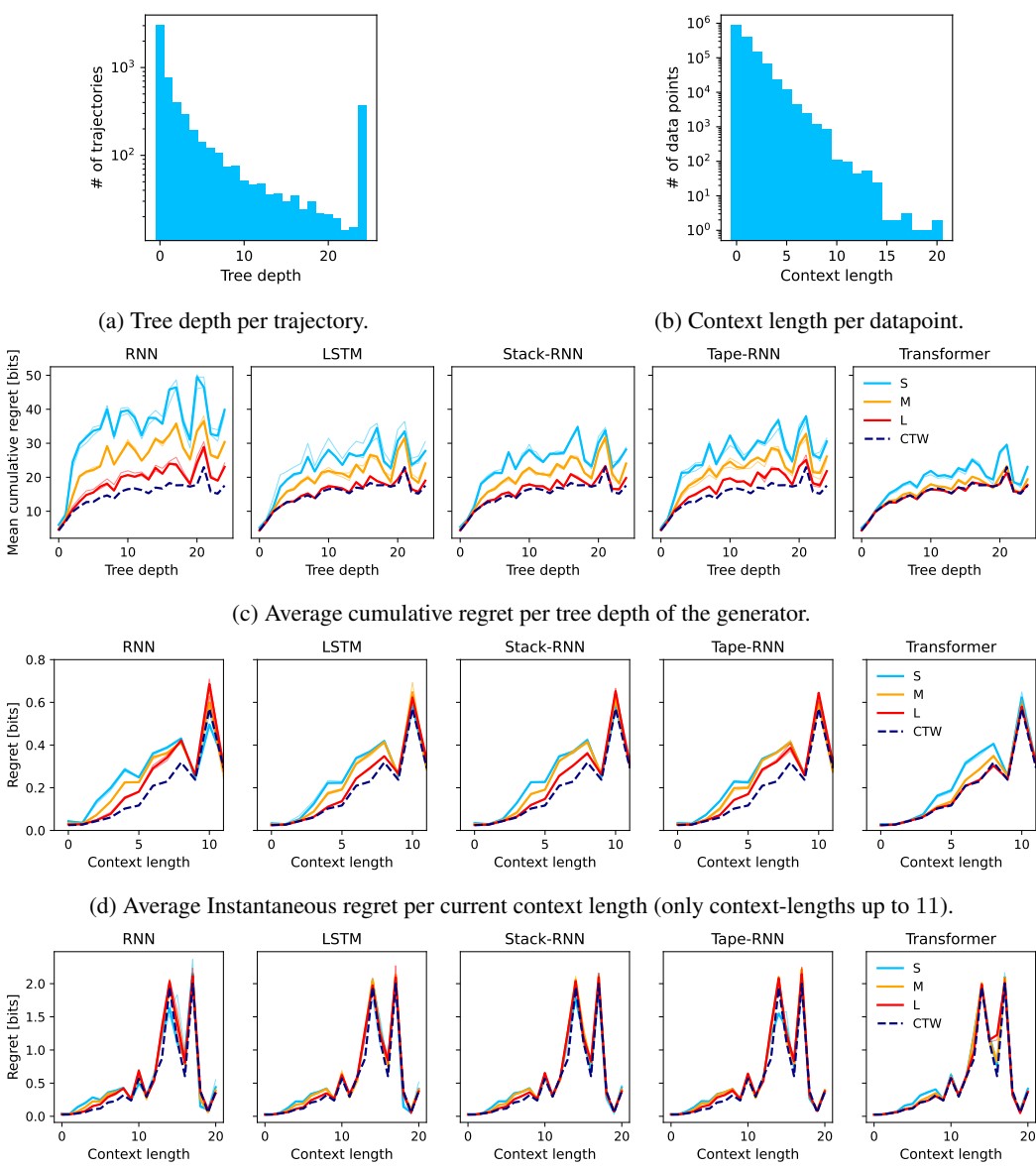

(a) Tree depth per trajectory.

(b) Context length per datapoint.

(c) Average cumulative regret per tree depth of the generator.

(d) Average Instantaneous regret per current context length (only context-lengths up to 11).

(e) Average Instantaneous regret per current context length (all context-lenghts).

Figure 6: Detailed results for the same 6k sequences as in Figure 1. Top two panels show histograms over tree depth (for all trajectories) and current context length (over all datapoints of all trajectories) use for evaluation in Figure 1. As expected, most generated trees have low depth and most datapoints have short contexts. The three lower panels show average cumulative regret per tree depth, and average instantaneous regret per context length respectively. Thin lines correspond to individual models (with different random initialization), bold lines show the median per model size. Across architectures smaller models only predict well for very short tree depth or very short context lengths (the maximum context length is upper bounded by the tree depth, but many contexts are much shorter than the maximum tree depth). Context lenghts $\geq 11$ are rare, which makes quantitative results in this regime less reliable.

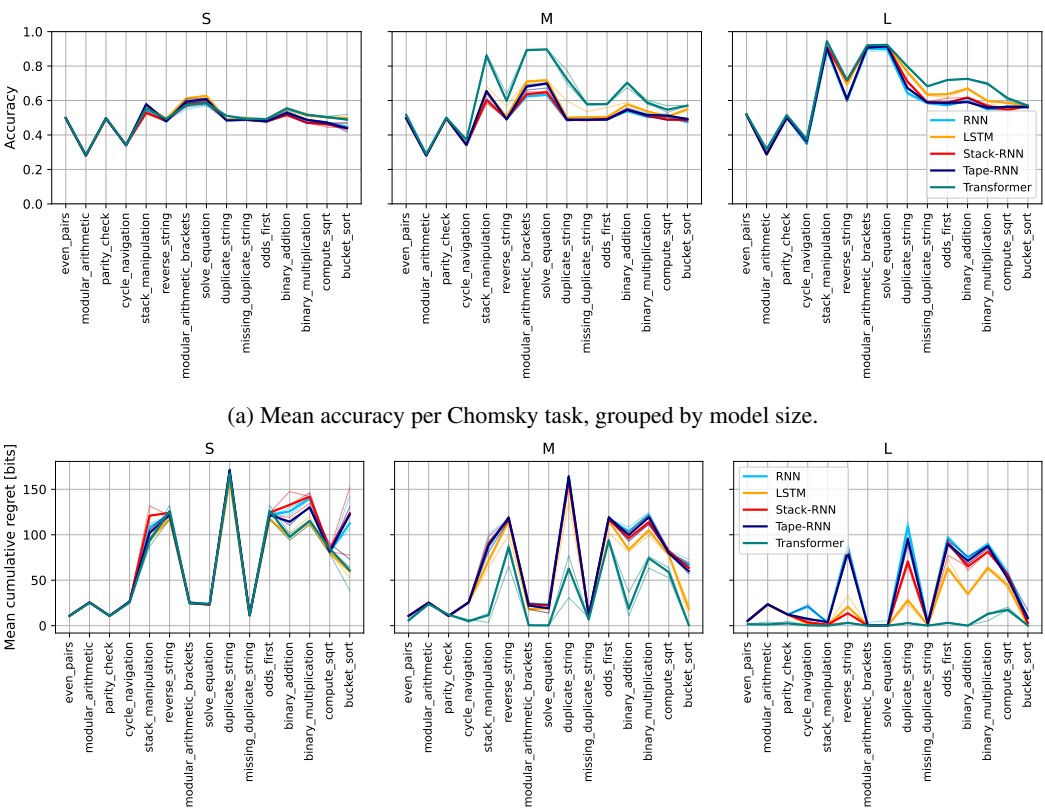

(a) Mean accuracy per Chomsky task, grouped by model size.

(b) Mean cumulative regret per Chomsky task, grouped by model size.

Figure 7: Detailed performance of networks trained and evaluated on the Chomsky tasks (6k sequences, 400 sequences per task; main results shown in Figure 2). Thin lines correspond to a single random initialization of a model, bolt lines show the median respectively.

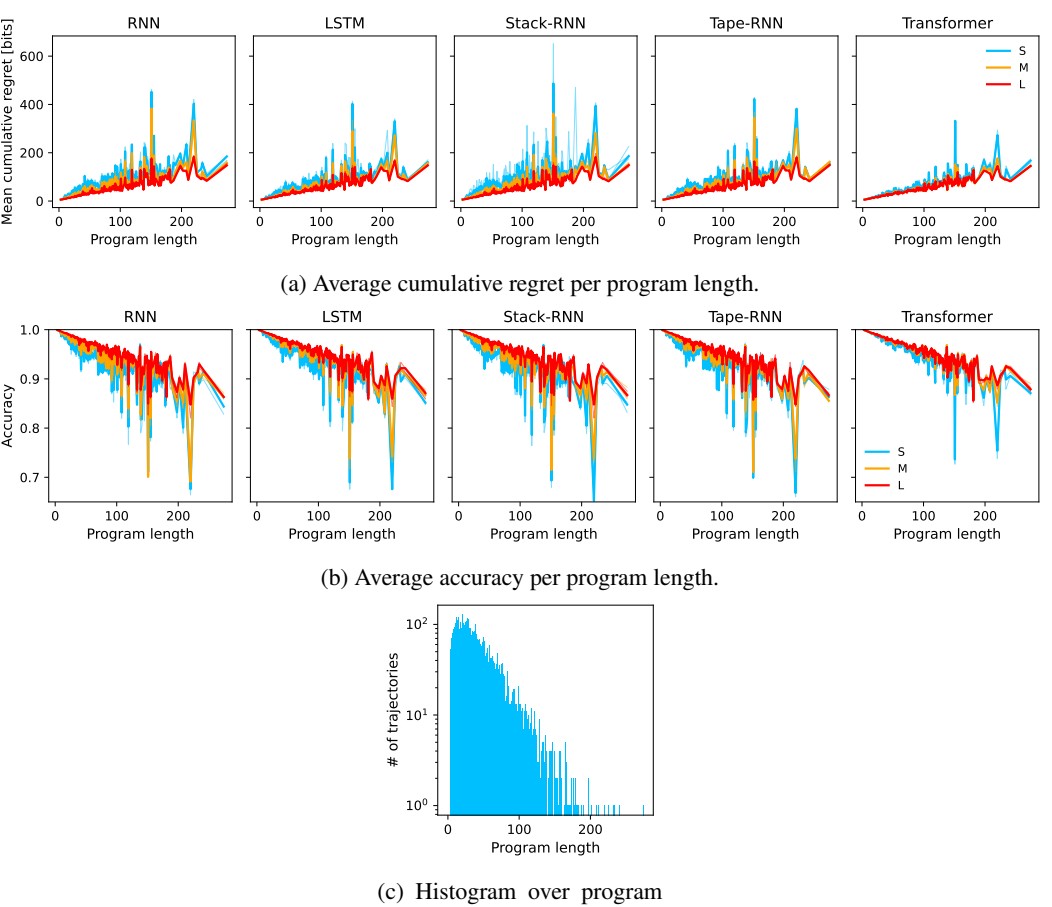

(a) Average cumulative regret per program length.

(b) Average accuracy per program length.

(c) Histogram over program lengths.

Figure 8: Results per program length for UTM in-distribution evaluation (same data as in Figure 3; 6k sequences, length 256).

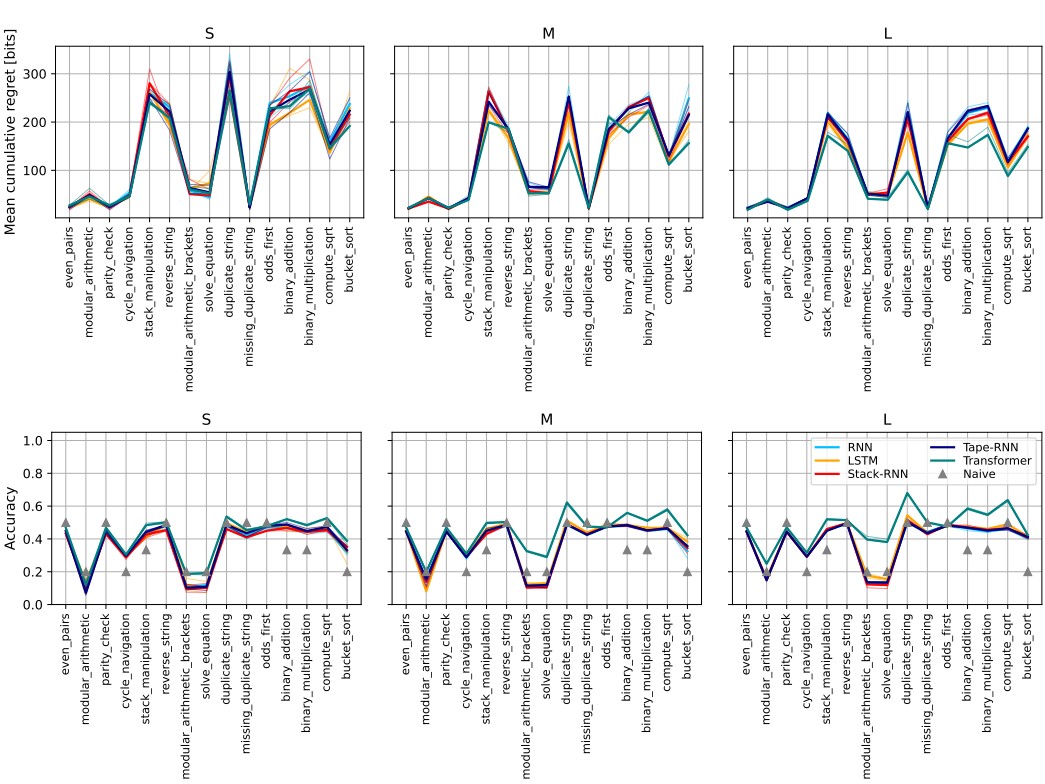

Figure 9: UTM transfer to Chomsky tasks.

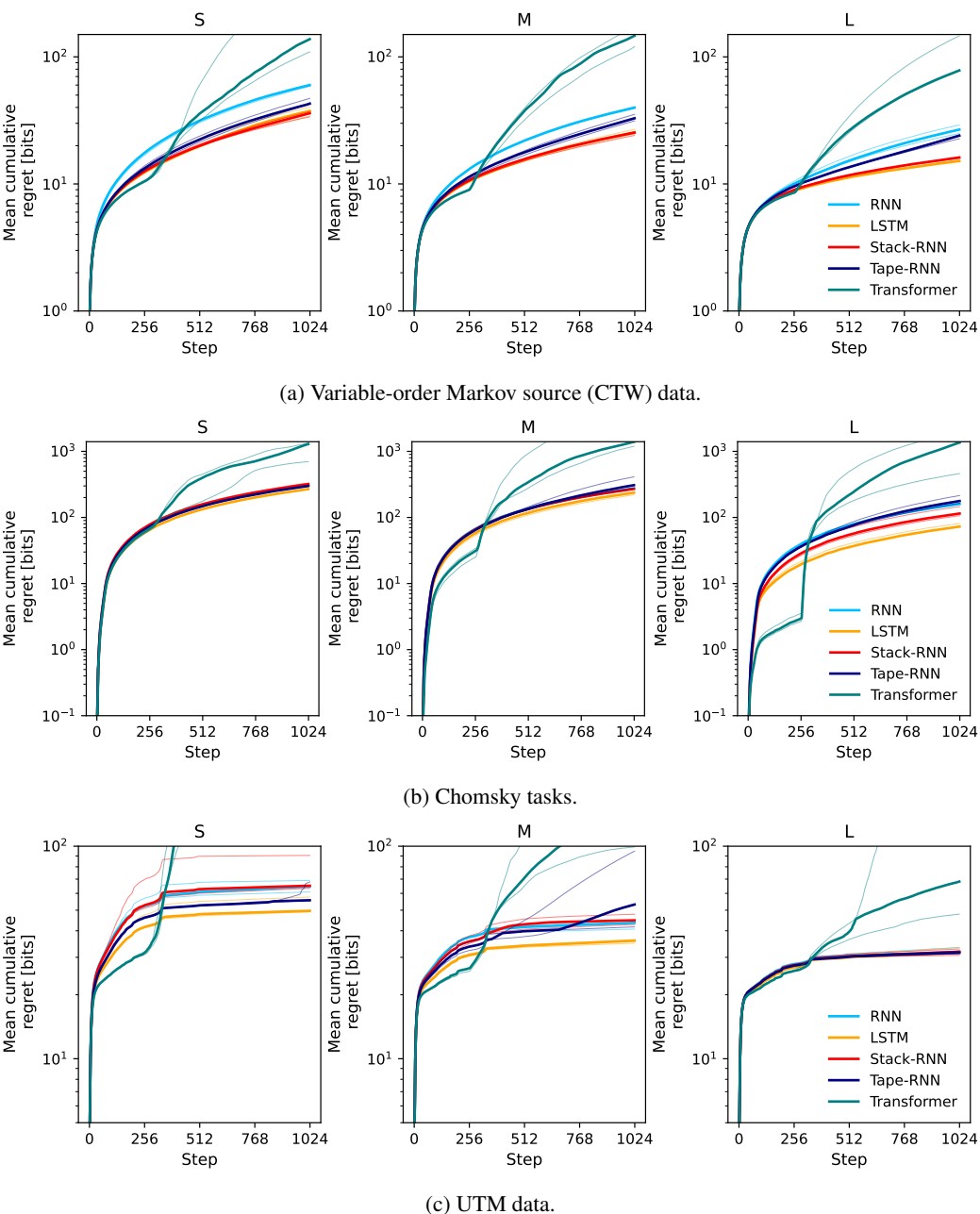

(a) Variable-order Markov source (CTW) data.

(b) Chomsky tasks.

(c) UTM data.

Figure 10: Full details of sequence-length generalization results. Models were trained on sequences of length 256 on their respective tasks, and are evaluated on 6k sequences of length 1024 from the same data generator type. Thin lines show individual models, bold lines are the median across random initializations of the same model. As expected, all models perform fairly well up to their trained sequence length, and then performance deteriorates more or less sharply. Most notably, prediction performance of the transformer models, regardless of their size, degrades very rapidly after step 256 and is often an order of magnitude worse than the other models. Across all experiments, LSTMs perform best in terms of generalizing to longer sequences.

