# OpenReview forum: "Neural Networks and Solomonoff Induction"
_ICLR.cc/2024/Conference — Submitted to ICLR 2024_

### Official Review · Reviewer_YTUE · 2023-11-01

**Soundness:** 3 good
**Presentation:** 1 poor
**Contribution:** 2 fair
**Rating:** 3
**Confidence:** 3

**Summary:**

- This paper investigates "amortizing Solomonoff Prediction into a neural network".
- They then "introduce a generalized Solomonoff prior".
- Lastly, they conduct experiments to show predictive performance on sequence prediction tasks, where they use meta learning with log-loss on a heterogeneous set of string related tasks.

**Strengths:**

The paper draws connections between Solomonoff Prediction and meta learning.

The paper seems to have a formal grasp on some concepts in computational complexity that are useful formalism for describing tasks in machine learning, for example ranking them according to the Chomsky hierarchy (does the task require a stack to solve? or a more complicated data structure)

**Weaknesses:**

Overall, the paper is hard for me to follow. In summary the issues are:

- Many definitions given up front (up to beginning of page 4) are fairly non-standard, given the general body of work that shows up at ICLR. At the same time, the presentation features little discussion of definitions after they are given, with most details relegated to appendix.


- Theorem statements in main text contain uncommon terms "probability gap" without definition.

- Definitions that are given contain other undefined terms within definition, e.g.:
  - An algorithmic data generating source µ is simply a computable data source by" A "data source" was not defined.
  - SI: Inductive inference aims to find a universally valid approximation to µ. What's "universally valid"?

- Propositions (e.g. prop 4, prop 8) are given and followed immediately  by a next section with no concluding sentence on what the takeaway should be or what the theorem means in words.

- Many data details (e.g."Variable-order Markov Source", one of the 3 experiments) are not defined in main text and details are relegated to appendix, and, as mentioned, are generally not particularly well known within the ICLR community.

- Many baselines / models not defined in main text: Stack-RNNs, Tape-RNNs,  Context Tree Weighting, where the last one is used as the main baseline.

- Important experimental details that are glossed over, e.g. there is a test distribution described as "out-of-distribution" in passing in the analysis of results without a formal experimental setup given for precisely what the shift between in- and out-distribution is.


I will be glad to raise my score if a major rewrite of this paper is undertaken. In particular it must be readable to wider audience without having to refer to the appendix for interpretation of main contributions or for understanding basic setup like datasets and baselines. As mentioned in "Questions" below, it is also necessary to clarify whether the experimental results are something distinct from running basic meta-learning on existing datasets. If not, is the significance in the connection to the theoretical results? If so, what is that connection?

**Questions:**

- It is stated that $\pi_\theta$  approximates the predictive distribution for each task $p(x_{t+1}|x_{\leq t}, \tau)$ for each task $\tau$ . However $\pi_\theta(x_{\leq t})$ is not notated to be a function of $\tau$. If $\pi_\theta$ is optimized with log loss it will learn a mixture of the predictive distribution across tasks rather than each task, unless extra assumptions are stated, such as that the support of $x_{\leq t}$ is disjoint across tasks for each $t$. Could the authors clarify  whether $\pi_\theta$ is also a function of $\tau$, and if not, what are the assumptions on the data that make this statement true?


- "Out-of-distribution" appears twice in the main text, including in the qualification of a test distribution. However, no particular definition of what is in versus out, or what the distribution shift is precisely, was given. A few sentences later, length-generalization is mentioned in passing, so I had to infer what in versus out meant. Usually it's really important to mention training versus test distribution details up front rather than in passing in the results. Could you please explain the precise experimental setup including data generation in more detail, in the main text?


- Finally, experimentally, it's not clear that the experiments run were anything different than running log likelihood optimization on a mixture of datasets. What's the practical/algorithmic difference or significance in what was run, and what should the takeaways be? If there is no difference, is the significance in the connection to the theoretical results? If so, what is that connection?

---

> ### Author Response · Authors · 2023-11-23
>
> Thank you for your insightful comments about our paper. We streamlined the paper thanks to your comments making it more readable and understandable. In particular, we have added to the revised manuscript:
>
> * We added more information on why some of our definitions are needed.
> * What we mean by “probability gap”: Let $\mu(x)$ be the probability that an (in)finite sequence \emph{starts} with $x$.  While probability distributions satisfy $\sum_{a\in\mathcal{X}}\mu(xa) = \mu(x)$, semimeasures exhibit \emph{probability gaps} and only satisfy  $\sum_{a\in\mathcal{X}}\mu(xa)\leq\mu(x)$.
>
> * We added what we mean by “data source”, which is simply an object that generates data, and replaced  “universally valid approximation to \mu” with a better description of Solomonoff Induction.
> * We added a Summary to Section 3 explaining why Propositions 4, 6 and 8 are useful.
> * Added more intuition to the Variable Order Markov Source in the main manuscript and in the appendix. See response to reviewer Mfzf for an expanded intuition for the variable-order Markov source.
> * We added a sentence explaining how Stack-rnn and Tape-rnn are just RNNs augmented with a stack and tape memory.
> * We do have a definition of what we mean by in- and out-of-distribution in the “Evaluation procedure” paragraph of Section 4. In the revised version we emphasize it better.
> * We respond to all your questions below.
>
> **“[...] Could the authors clarify whether $\pi_\theta$ is also a function of $\tau$, and if not, what are the assumptions on the data that make this statement true?”**
>
> $\pi$ has no dependency on $\tau$. In addition, the aim of $\pi$  is to approximate the mixture distribution (over $\sum_\tau p(x | \tau)p(\tau)$). Since this is fairly standard we talked about this in the Background section on Meta-learning.
>
> **“Out-of-distribution" appears twice in the main text, including in the qualification of a test distribution.[…] Usually it's really important to mention training versus test distribution details [...] Could you please explain the precise experimental setup including data generation in more detail, in the main text?”**
>
> Our definition of in- and out-of-distribution is located on “Evaluation procedure” in Section 4. We highlighted this better.
> We added more intuition and detail  in the main text about the experimental setting.
>
> **“Finally, experimentally, it's not clear that the experiments run were anything different than running log likelihood optimization on a mixture of datasets. What's the practical/algorithmic difference or significance in what was run, and what should the takeaways be? If there is no difference, is the significance in the connection to the theoretical results? If so, what is that connection?”**
>
> Thanks for the comment, this is exactly the point that we are trying to make. That is, learning from the log-loss on a distribution of tasks leads to an approximation of Solomonoff Induction in a similar fashion as Ortega et al 2019, but only when the “data-mixture” is of a particular form, generated by UTMs, as described by our Section 3. This is exactly the theoretical takeaway. Furthermore, Thm 9 allows for more flexibility in generating this type of data, in which changing the distribution over programs may unblock generating the most important data quicker, which could lead to better performance. We use this fact in our experiments to generate better outputs while maintaining universality.  On the experimental side, we show mainly two things: 1) some networks are capable of achieving close to the Bayes-optimal solution for data sources that involve a mixture over programs (in this case Variable-Order  Markov Data is generated by mixing trees which can be effectively thought of as programs). 2) Training networks on UTM data exhibits transfer to the other tasks we considered. This is a hint that the UTM data exhibits rich transferable patterns. We believe that designing better UTMs or distribution over programs might lead to even more transfer to more complex domains.

---

### Official Review · Reviewer_iUvX · 2023-11-03

**Soundness:** 3 good
**Presentation:** 3 good
**Contribution:** 2 fair
**Rating:** 5
**Confidence:** 5

**Summary:**

In this work, authors theoretically investigate how universal approximators using a dataset converge to SP in some limit and show that universality is maintained even when underlined distribution shifts. They experiment with Transformers and LSTM NNs to show model complexity increases with increase in parameters, such that convergences can be seen on challenging dataset.

**Strengths:**

1. The paper is well written
2. Good set of experiments

**Weaknesses:**

1. Novelty is limited
2. Several key papers are not cited

Below, I provide my detailed review.

**Questions:**

It is true only for first-order RNNs that they are Turing complete with infinite precision and time, however, tensor RNNs with and without memory are shown to be equivalent to TM with finite precision [5,6] and also UTM

Lemma 10, Corollary 11, Lemma 12-14 [ 1-2] – Theorem 9 in paper shares some similarities, slightly different ways to prove the same property.

Theorem 3 Linear separation [4] – the paper could benefit by showing how various layers in transformers cause linear separation using hard attention and would lie in Banach space dual. Again, with some assumption, it is trivial to show how their approach is also universal.

Generalized Solomononff semimeasures definition and Theorem 9 in the paper also share similarity with [3]; second majority of suggestions and claims are given in [3]. Furthermore they have shown some experiments and multiple hypothesis generation can be seen as a case of meta-learning. There are several lemmas on recursive functions, that can be extended with modern RNNs such as LSTM and even for Transformers (assuming within a finite length, they approximate RNN).

Authors should cite these line of work. Thus it seems the current manuscript is more incremental aligned with the experimental setup in the meta-learning space using modern NNs.

Finally, it is hard for me to see what values the current method provides to the community; I will briefly discuss why I feel this,

* theorem 11 in [7] proves that equivalence between two RNN is undecidable, Theorem 6 shows that consistency problem in RNN is also undecidable, Theorem 7 shows 2 layer RNN using BPTT on a finite corpus is necessary not consistent, furthermore Theorem 11 and 8 points out best string problem is NP-hard and in some cases undecidable. Given we know above properties for RNN, that is also true for transformers with some conditions, thus I am not sure Solomonoff induction would help in getting universal capability of the modern day NNs

* Second RNN and transformers are turing complete comes from a unrealistic assumptions where entire tape is encoded into a tape. Based on bignum arithmetic we can see there is infinitely many hierarchies across various natural numbers, and works in infinite space. Therefore, what practical benefits it offers is still a open question.

* Third when we move to UTM space and show RNN is equivalent to UTM will also work in infinite space and time

* Fourth Solomonoff induction also requires infinite samples, given everything or in simple words all components are working in infinite space, how can one show practical universality? Nor can it be claimed that the model trained on the dataset is universal. So, I would advise authors to lower down the claim as it is highly misleading.


It would benefit if authors can provide insight how transformers and RNNs LM can benefit. For instance, by showing how they work when state space is small vs large, symbols are increased, model is trained on short strings and tested on longer. How do attention weights attend in such scenarios, how does LSTM memory adapt to these changes? Do you observe any tape-like or even stack-like behaviour etc. showing these analyses would further benefit the paper and will help understand how using SI can help LLMs reason about the world in some finite space.



1.	Sterkenburg, T.F., 2017. A generalized characterization of algorithmic probability. Theory of Computing Systems, 61, pp.1337-1352.

2.	Wood, I., Sunehag, P. and Hutter, M., 2013. (Non-) equivalence of universal priors. In Algorithmic Probability and Friends. Bayesian Prediction and Artificial Intelligence: Papers from the Ray Solomonoff 85th Memorial Conference, Melbourne, VIC, Australia, November 30–December 2, 2011 (pp. 417-425). Springer Berlin Heidelberg.

3.	Li, M. and Vitanyi, P.M., 1992. Inductive reasoning and Kolmogorov complexity. Journal of Computer and System Sciences, 44(2), pp.343-384.

4.	Sunehag, P. and Hutter, M., 2013. Principles of Solomonoff induction and AIXI. In Algorithmic Probability and Friends. Bayesian Prediction and Artificial Intelligence: Papers from the Ray Solomonoff 85th Memorial Conference, Melbourne, VIC, Australia, November 30–December 2, 2011 (pp. 386-398). Springer Berlin Heidelberg.

5.	Stogin, J., Mali, A. and Giles, C.L., 2020. A provably stable neural network Turing Machine. arXiv preprint arXiv:2006.03651.

6.	Mali, A., Ororbia, A., Kifer, D. and Giles, L., 2023. On the Computational Complexity and Formal Hierarchy of Second Order Recurrent Neural Networks. arXiv preprint arXiv:2309.14691.

7.	Chen, Y., Gilroy, S., Maletti, A., May, J. and Knight, K., 2017. Recurrent neural networks as weighted language recognizers. arXiv preprint arXiv:1711.05408.

---

> ### Author Response · Authors · 2023-11-23
> **Part (1/2)**
>
> We thank the reviewer for the insightful comments and references. We cited all suggested papers, added a limitation section in the Discussion and highlighted better our novel contributions.
>
> **“... tensor RNNs with and without memory are shown to be equivalent to TM with finite precision [5,6]...”**
> Thank you for the comment; we will add these references to our manuscript.
>
> **“Lemma 10, Corollary 11, Lemma 12-14 [ 1-2] – Theorem 9 in paper shares some similarities …”**
> and
> **“Generalized Solomononff semimeasures definition and Theorem 9 in the paper also share similarity with [3]...”**
>
> Thank you for the references. While references 2 and 3 indeed talk about basic and fundamental ideas around Solomonoff Induction they only contain the statement that M is universal, they do not contain any discussion of M^Q (the non uniform case). With respect to reference 1, which we were unaware of, indeed shows similarity to our Thm 9. This means that the conclusions obtained by our Thm 9 are not novel. It seems we have independently discovered similar conclusions to reference 1, though our proof seems to be more self-contained and short when compared to ref 1.  We have adapted our claims in the revised version of the manuscript stating the above.
>
>
> **Theorem 3 Linear separation [4] – the paper could benefit by showing…**
> This would definitely be an interesting element to investigate to better understand transformers, however, we think it falls outside the scope of this paper.
>
> **“Authors should cite these line of work. Thus it seems the current manuscript is more incremental aligned with the experimental setup in the meta-learning space using modern NNs.”**
> We added all citations since, indeed, they are related to our work. It is true that a major part of our work is to experimentally try to get closer to Solomonoff Induction by meta-learning neural networks (see our general response for an argument why we believe the empirical work is important). We think that this is aligned with ICLRs audience and topics. We also believe the ICLR audience would greatly benefit from thinking more about the links between empirical ML and the strong theory of Solomonoff Induction, which is one of the subgoals of this paper.
>
> **“theorem 11 in [7] proves that equivalence between two RNN is undecidable […] Given we know above properties for RNN […]  I am not sure Solomonoff induction would help in getting universal capability of the modern day NNs”**
> If the reviewer’s comment refers to the fact that a non-universal model could fail in approximating SI when training on our universal data, we agree that this is the case and it is not a surprise (i.e. non-realizability). If, instead, the reviewer means that this could also happen even if we have universal architectures, we are not so sure about this. The mentioned properties for RNNs (useful citations we add to the revised version), can possibly translate to transformers, however, we note that our work simply takes current neural networks architectures and tests them empirically on various data sources including UTMs. In this work, we are not concerned about how to develop easily-trainable universal architectures as we mention in the introduction, which would be a major breakthrough and a paper on its own. We note that while neural network universality is necessary for full Solomonoff Induction, universality won’t help with the fact that Solomonoff Induction is uncomputable/undecidable.  Nevertheless, more compute and bigger universal architectures should help when aiming at approximating Solomonoff Induction (that is computable/decidable for bounded time/space versions). This is the aim of our paper, and in line with our experiments, we show that increasing neural network size leads to better performance and transfer.

---

> ### Author Response · Authors · 2023-11-23
> **Part (2/2)**
>
> **“Second RNN and transformers are turing complete comes from a unrealistic assumptions [...]“**
>
> We wholeheartedly agree with the reviewer. Unfortunately, there is no universally accepted universal architecture that practitioners use effectively without unrealistic assumptions. In fact, we believe this is an actual open problem and not the focus of our paper. We are simply concerned with “how good widely used architectures (rnns, lstms transformers etc) are when approximating our data generating sources?”.
>
> **“Third when we move to UTM space and show RNN is equivalent to UTM will also work in infinite space and time.”**
> We agree that this could be the case.
>
> **“Fourth Solomonoff induction also requires infinite samples [...]”**
> This is an interesting question. We agree that our way of approximating Solomonoff Induction requires infinite samples, and that even if we would have a universal architecture (realizability) which can be easily trained (learnability), one can  we can never guarantee that  any approach (including ours) is exactly mimicking the universality of a Solomonoff Inductor (since it is uncomputable). However, our theory shows that in the limit this universality should be achieved. We also consider the finite space case in Definition 5 that uses limited resources (bounded time and space). We added a “Limitations” section explaining these nuances. Note that, we are not aware of any practical work that generates samples directly from a UTM according to our Definition 5, and then uses these samples to learn a predictor. So, our way of approximating Solomonoff Induction is novel, as far as we can tell.
>
>
> **“It would benefit if authors can provide insight how transformers and RNNs LM can benefit. For instance, by showing how they work when state space is small vs large, symbols are increased,**
>
>
> We conducted experiments with small and large state spaces (our architectures range from small , medium and large sizes). W.r.t. Alphabet sizes we opted for a fixed alphabet size of 17 for most tasks since this allows comparing transfer performance across tasks as we do in our experiments. When having small alphabet sizes, certain computations might require longer sequences when compared to the same setting with large alphabet sizes, this is because large alphabets in a sense have more bandwidth.
>
> **“model is trained on short strings and tested on longer. “**
> We do have these experiments for all tasks, see Figures 1,2 and 3 right panels. Basically, our conclusions are that LSTMs do very well when tested on longer sequences, whereas transformers fail to do so.
>
> **How do attention weights attend in such scenarios, how does LSTM memory adapt to these changes? Do you observe any tape-like or even stack-like behaviour [...]”**
> These are interesting questions, but they fall beyond the scope of this paper, and we refer the reviewer to (Deletang et al 2022) where there is an exploration of such kinds of questions. In summary, given that RNNs cannot solve the tasks that a Stack-RNN or Tape-RNN can solve, this means that they are likely not implementing a stack or a tape under the hood in their internal activations.

---

### Official Review · Reviewer_Mfzf · 2023-11-08

**Soundness:** 3 good
**Presentation:** 3 good
**Contribution:** 3 good
**Rating:** 5
**Confidence:** 2

**Summary:**

The authors explore approximate versions of Solmonoff induction via meta-learning and neural networks.  Their experimental setup compares the performance of a variety of deep neural network architectures within their framework on several algorithmically generated data sets.

**Strengths:**

The paper is clearly written and motivates the general problem.

The discussion, although it is primarily focused on Turing machines, is relevant broadly to themes in modern ML, e.g., large language models and other increasing large DNNs trained on massive data sets.  As a result, this work might help make connections between more classical AI and modern ML.

**Weaknesses:**

While the discussion is clear in many places, it also assumes quite a bit a background without references, e.g., "Kolmogorov's probability axioms".  As this is a submission to an ML conference, I suggest that the authors provide the necessary context to aid unfamiliar readers.  In the same vein, not many participants at ICLR are likely to be familiar with Solmonoff induction. So, the fit might be better at a more traditional AI venue -- the advances here are more from about using existing NN tools rather than pushing the state-of-the-art in deep NNs.

The experimental setup is missing some details, e.g., how many training examples are there?

**Questions:**

- I don't really have good intuition about how varied the prediction tasks are.  Can you provide a bit more intuition here?

- Like large language models, I expected that you would need a significant amount of data for training in this case.  Can you talk a bit more about data sizes, and why the results are or are not expected?

---

> ### Author Response · Authors · 2023-11-23
>
> We thank the reviewer for the helpful comments and positive feedback. Below we provide detailed responses.
>
> **“... I suggest that the authors provide the necessary context to aid unfamiliar readers…”** :
> We rephrased the description of Semimeasures to be more accessible to readers.. We also provide further explanations about Solomonoff Induction to be more easily understood by ML practitioners. We emphasize in the introduction that the focus of the paper is not to develop a novel architecture but to study more empirical versions of Solomonoff Induction using modern neural networks.
>
> **“... The experimental setup is missing some details, e.g., how many training examples are there?..”**:  In the Neural Predictors section we describe how we used 500k iterations with a batch size of 128. Since we use data generators (not fixed datasets) we effectively generated 500k*128 sequences of various lengths depending on the experiments. We are happy to clarify other details that the reviewer thinks are missing.
>
> **“I don't really have good intuition about how varied the prediction tasks are. Can you provide a bit more intuition here?”:**
> We added intuitions in the revised version of the manuscript for all tasks, see below.
>
> * The Chomsky tasks are tasks that manipulate strings, for example, reversing an input string, or computing modular arithmetic operations. More details n our Appendix D3.
>
> * For the UTM (BrainF*ck/BrainPhoque) tasks we generate random programs (that can effectively encode any structured sequence generation process) and run them in our UTM and get the outputs. For example, in principle a program could generate the image of a cow, a chess program, or the books of Shakespeare. But of course these programs are extremely unlikely to be sampled. In practice, the generated programs are short and their outputs bear more resemblance with regular or context-free languages, which explains partly why transfer to the Chomsky tasks is possible at all. See Figure 5 in the Appendix for some examples.
>
> * A Markov model of order k sequentially assigns probabilities to a string of characters by looking, at step t in the sequence, at the suffix string from t-k to t. This suffix is used to lookup the model parameters to make a prediction of the next character. A variable Markov model (VMM) is a Markov model where the value of k can vary depending on the suffix. A VMM makes its prediction using a suffix tree. CTW is a variable Markov model predictor that performs Bayesian inference efficiently over all possible suffix trees as it reads and predicts the sequence. Additionally, any predictor (such as CTW) can be used as a generator, by sampling sequentially from the predictive probabilities.
> Lastly, the task on variable-order Markov sources considers data generated from tree structures. For example, given the binary tree
> ```
>          Root
>   0/                \1
> Leaf_0       Node
> 		0/         \1
>         Leaf_10       Leaf_11
> ```
>
> And given the history of data “011“ (where 0 is the first observed datum and 1 is the last one) the next sample uses Leaf_11 (because the last two data points in history were 11) to draw the next datum using a sample from a Beta distribution with parameter Leaf_11. Say we sample a 0, thus history is then transformed into “0110” and Leaf_10 will be used to sample the next datum (because now the last two datapoints that conform to a leaf are 10), and so forth.  This way of generating data is very general and can produce many interesting patterns ranging from simple regular patterns like  01010101 or more complex ones that can have stochastic samples in it. Larger trees can encode very complex patterns indeed. For more information about the way we generate this data you can check the Appendix D2.
>
> **… Can you talk a bit more about data sizes, and why the results are or are not expected?”**
> We train for 500k iterations with batches of 128 sequences of length 256 in most experiments. That means we use around 16 Billion “tokens” to train our models. Large language models are trained using datasets sizes of the order of Trillions of tokens e.g. Mistral 7B uses about 8 Trillion tokens. Given this context, our models could still be trained further if we make them bigger, specially in the case of UTM data where we can generate at will any number of sequences. Our results on Variable Order Markov Sources are novel and somewhat reassuring since no previous work has shown the capabilities of neural networks to reach Bayes-optimal performance on such type of algorithmic universal data. The results on UTM data are a positive surprise since they demonstrate our hypothesis that practical UTM data contains useful transferable patterns to improve performance on other tasks. What we expect is that using better UTMs and training on the order of Trillions of tokens could significantly improve transfer to more “real-world” tasks including vision and language.

---

### Author Response · Authors · 2023-11-23
**General response**

We thank the reviewers for their helpful comments. We are pleased to see that the paper is well written and motivated (Mfzf, iUvX), that we conducted a good set of experiments (iUvX) and that we provide useful connections between Solomonoff Induction and meta-learning relevant for the ML, classical AI and Deep Learning community (YTUE, Mfzf). We make a general response here for all reviewers and also provide detailed responses to each reviewer individually under their reviews.

The two main points to address appear to be about the necessary background to understand the paper (Mfzf, YTUE) and its positioning with respect to other related works (iUvX). We address both points in the new revision by rewriting parts of the paper to be more accessible to the ICLR audience and comparing to the literature.

We have revised the manuscript to be more accessible to readers without a background in information theory and the theory of computation. Both topics are part of most undergraduate curricula in computer science and ML, but we agree with the reviewers that the topics have lately received less attention in the neural networks community. Since the conference-paper format has a strict page limit, we must balance the depth of the background material in the main paper with novel and original content (both theory and experiments), making it necessary to defer some details to the appendix. We believe that with the revised version we have struck a better balance between background/textbook material and original content by moving some of the more detailed results and arguments into the appendix. Since accessibility by the wider ICLR audience is one of our main goals, we would be keen to hear from reviewers whether they think that there are any open accessibility issues.

Regarding the positioning of the paper, we thank the reviewers for highlighting some missing references related to our work. Consequently, we have added all citations to the revised manuscript and better explained the relationship and differences between the suggested works and our work. In particular, it seems we have independently discovered similar results as in reference 1 from reviewer iUvX, while our proof of Thm 9 is shorter and more self-contained. We adapted our claims in the new revision of the manuscript.  We appreciate the theoretical results and discussion pointed out by iUvX. Unfortunately, when dealing with neural networks in practice, (positive) theoretical results regarding representability and realizability are often somewhat vacuous. The important additional question in practice is whether mini-bach based SGD is actually able to reliably find parameter sets to realize certain solutions. As Deletang et al. (cited in the paper) demonstrate, this is often not the case, even for simple computational tasks and modern SOTA architectures and training procedures. We thus believe that our theoretical study must crucially be accompanied by some empirical investigation, which paints a more nuanced picture and allows to draw more practical conclusions. To the best of our knowledge our paper is the first paper to  empirically investigate learning neural approximations to Solomonoff induction  by generating synthetic data from UTMs or variable-order Markov sources and using modern neural architectures and training setups. Our theoretical results establish that good approximations to Solomonoff induction are obtainable in principle via meta-training, our empirical results show that is possible to make practical progress in that direction, but also that many questions remain open (e.g., how to construct efficient datasets for meta-learning, and that today’s architectures are not universal in practice). In addition, we are not aware of  other work that shows how training on universal data sources (like UTMs) improves performance on other tasks. Indeed, we show how our networks trained on UTM-generated data transfer to Chomsky and CTW tasks, meaning that the data generated from the UTM contains sufficiently rich patterns transferable to the aforementioned tasks. Given this, we think our work is general, novel and of interest for the ML community given its empirical contributions and its ties with pure theory.

---

### Meta-Review · Area_Chair_dRMG · 2023-12-13

**Metareview:**

The paper revisits Solmonoff induction and creates an alternative approach for approximating the SI solution based on meta-training of neural networks. All of the reviewers on this paper were negative. From the less expert reviewers, there was a strong push to make the paper more accessible especially with respect to the definitions and background at the start of the paper. From more expert reviewers, there was a push towards better contextualization of the work with respected to existing related work.

**Justification For Why Not Higher Score:**

Expert reviewers thought the contribution with respect to existing work was unclear. Less confident reviewers wanted a much more accessible background.

**Justification For Why Not Lower Score:**

N/A

---

### Decision · Program_Chairs · 2024-01-16

Reject